# Osmotic disruption of chromatin induces Topoisomerase 2 activity at sites of transcriptional stress

William H. Gittens ✉, Rachal M. Allison, Ellie M. Wright,
George G. B. Brown & Matthew J. Neale ✉

Transcription generates superhelical stress in DNA that poses problems for genome stability, but determining when and where such stress arises within chromosomes is challenging. Here, using G1-arrested *S. cerevisiae* cells, and employing rapid fixation and ultra-sensitive enrichment, we utilise the physiological activity of endogenous topoisomerase 2 (Top2) as a probe of transcription-induced superhelicity. We demonstrate that Top2 activity is surprisingly uncorrelated with transcriptional activity, suggesting that superhelical stress is obscured from Top2 within chromatin in vivo. We test this idea using osmotic perturbation—a treatment that transiently destabilises chromatin in vivo—revealing that Top2 activity redistributes within sub-minute timescales into broad zones patterned by long genes, convergent gene arrays, and transposon elements—and also by acute transcriptional induction. We propose that latent superhelical stress is normally absorbed by the intrinsic topological buffering capacity of chromatin, helping to avoid spurious topoisomerase activity arising within the essential coding regions of the genome.

In order to package genomes within the tight confines of a nucleus, genetic material must be compacted via higher-order levels of chromatinisation, but in an organised manner that facilitates the essential processes of DNA replication, chromosome segregation, and the ubiquitous activity of transcription by RNA polymerases. In all cases, these processes cause topological changes within the underlying helical double-stranded DNA substrate, yet characterising where and when such changes arise in vivo remains a challenge due to limitations in methodological techniques able to directly probe DNA topological state in situ.

DNA is overwound ahead of, and underwound behind, transcribing RNA polymerases, within a paradigm known as the twin-domain model of superhelicity[1–3]. Whilst potentially important for regulation[4–6], the accumulation of superhelicity can pose an impediment to transcription[7]. Eukaryotic topoisomerases 1 and 2 (Top1 and Top2) regulate superhelical twist and writhe by generating transient DNA strand breaks, yet do so via distinct mechanisms[8]. Whilst Top1 acts as a swivelase by cleaving a single strand of DNA to release twist[9], Top2 instead relaxes writhe by targeting plectonemic DNA structures[10,11]—generating transient DNA double-strand breaks (DSBs) that allow gating of one helix through another. DNA twist and writhe are interconvertible[12], explaining the high functional redundancy of Top1 and Top2 in supporting transcription[13]. Notably, however, loss of Top2 (but not Top1) specifically decreases long gene transcripts by stalling RNA polymerase 2 (RNAP2), via a mechanism specifically linked to the formation of positive, rather than negative supercoils[14]. Thus, Top2, but not Top1, activity may be essential to support transcription in certain genomic contexts and cellular conditions. Understanding when and where superhelical topoisomerase substrates form will help to clarify the relationship between transcription, superhelicity and the functions of these fundamental and ubiquitous enzymes.

Notwithstanding the functional redundancy of Top1 and Top2, in vitro and theoretical work suggests that structural reorganisation within the chromatin fibre itself may buffer transcription-associated

Genome Damage and Stability Centre, School of Life Sciences, University of Sussex, Brighton, UK. ✉e-mail: W.Gittens@sussex.ac.uk; m.neale@sussex.ac.uk

superhelicity, bypassing some requirements for topoisomerase activity[15–22]. For example, molecular tweezer experiments demonstrate that nucleosome arrays can absorb a high degree of torsion without a great change in length—in contrast to naked DNA, which rapidly deforms upon torsion due to the formation of positive or negative plectonemes[15,16,18,19,22]. To explain these results, it is proposed that chromatin fibres fold into left-handed superhelical structures that are compliant—specifically—to positive torsion, with minimal increase in torque[16]. Such observations suggest that chromatin comprises a reservoir of negative superhelicity that can absorb positive torsion thereby aiding the progression of polymerases during transcription and DNA replication. Yet these models have been proposed based largely on in vitro single-molecule manipulation, and evidence in vivo is limited.

Here, we employ an updated CC-seq-v2 method to map the physiological catalytic activity of Top2 and draw insight into its relationship with local transcription. High temporal resolution, on the order of seconds, allows us to map transcription-dependent Top2 activity throughout an acute ionic disruption of chromatin via osmotic shock—revealing induced hotspots of activity that suggest that chromatin buffers positive topological stress in vivo.

## Results

### Mapping physiological Top2 activity in etoposide-unchallenged cells

To understand how superhelical stress is organised by genome structure and modulated by transcription, we mapped the transient sites of Top2 covalent complex (Top2cc) formation arising genome-wide during the natural catalytic cycle of Top2-mediated DNA strand passage on supercoils in G1-arrested *S. cerevisiae* cells, utilising rapid fixation and enhancements of a prior method[23] that greatly increases sensitivity and precision (Supplementary Fig. 1A–E, Methods). Major improvements to the method include (1) immediate fixation of the cells with −20 °C ethanol to instantaneously denature native physiological Top2ccs (Supplementary Fig. 1A) without prior stabilisation with etoposide, (2) extended ligation times to maximise DNA library yields from low inputs, and (3) a new dephosphorylation step to prevent adaptor ligation to free DNA ends (Supplementary Fig. 1B). Collectively, these modifications dramatically reduce non-specific noise in the maps, improving dynamic range (Supplementary Fig. 1C).

Whilst prior methods[23–25] have relied upon exposure of cells to the Top2 poison etoposide (VP16) to stabilise Top2cc, our updated methodology (CC-seq-v2) resolved strong strand-specific peaks throughout the genome even in untreated cells (Fig. 1A). Notably, peaks were completely lost upon degron-mediated Top2 depletion, demonstrating specificity for physiological Top2 activity (Fig. 1B). At fine scale (nucleotide-resolution), top and bottom-strand CC-seq signals are correlated (Pearson R = 0.65) when offset by 3 bp (Fig. 1C), as expected for the known structure of a Top2cc[26,27]—indicating that our methods independently isolate both sides of a Top2-DSB. Consistent with this, the average DNA-sequence composition around cleavage sites is rotationally symmetrical and lacks the non-physiological skews arising from etoposide-dependent poisoning of individual Top2 monomers[23] (Fig. 1D). Collectively, these observations demonstrate our ability to quantify physiological Top2 activity across the genome with nucleotide-resolution and strand specificity.

Comparing maps generated using CC-seq-v2 in untreated cells with those generated using our prior CC-seq-v1 method (that required etoposide exposure)[23], we observed broad similarities in the distribution of Top2ccs at whole-chromosome scales (Pearson R = 0.63) (Supplementary Fig. 2A), and at medium scale (Supplementary Fig. 2B–E). We also observed some differences in peak distributions and magnitudes—but these could be explained by differences between the cell cycle phase and/or transcriptional landscape inevitably produced by intrinsic CC-seq-v1 requirements (the use of a multidrug

sensitive *pdr1Δ* background, plus incubation for 4 h with etoposide) with those employed here for CC-seq-v2 (*PDR1+*, G1-arrest, no drug treatment). Similarly to VP16-stabilised Top2ccs mapped by CC-seq-v1, physiological Top2ccs mapped by CC-seq-v2 were enriched in promoter and terminator intergenic regions (IGRs), and depleted in gene bodies (Supplementary Fig. 2F–G). Indeed, this relationship was enhanced in CC-seq-v2, suggesting increased signal/noise, which was also evidenced by a broader distribution of signal densities genome-wide (Supplementary Fig. 2H) and by a higher correlation between top and bottom strand signals with a 3 bp relative offset (Supplementary Fig. 2I)—characteristic of Top2 DNA double-strand cleavage.

Confident in our ability to map Top2 activity without requiring etoposide exposure, we characterised in more detail the CC-seq V2 datasets generated in untreated cells. At medium scale (0.1–1 kb) peaks of Top2 activity are distributed in a manner that is highly influenced by gene structure and orientation (Fig. 1A), with strong enrichment in promoter-containing tandem and divergent intergenic regions (IGRs), followed by convergent IGRs (Fig. 1E)—and with strong anti-correlation with nucleosome occupancy (Fig. 1A, F). Yet, at a broad scale (>10 kb), the dynamic range of CC-seq was much lower than that of RNA-seq, indicating that Top2 activity is remarkably homogeneous across chromosomes (Fig. 1G–H), whereas transcriptional activity is not (Fig. 1G–H). Furthermore, despite much prior work implicating Top2 in the resolution of transcription-associated DNA supercoils[2,8,13,14], we were surprised to discover that distributions of global transcription and Top2 activity are completely uncorrelated across the chromosome (Pearson R = -0.01, P value = 0.70) (Fig. 1G).

### Multi-layered aspects of transcriptional organisation pattern genome-wide Top2 activity

Given the unexpected lack of correlation between physiological Top2 activity and transcription at the chromosome-wide scale, we sought to understand the relative organisation of these entities in more detail. We first focussed our analysis at the single gene scale. In aggregate, Top2 activity was enriched within the nucleosome-depleted region (NDR) of promoters, lower at terminators, and weak within genes (Fig. 2A). Fractionating promoters into recently defined classes[28] revealed that Top2 activity was highest at promoters unbound by all except the pre-initiation complex (UNB) (Fig. 2B–Ci). Comparatively, Top2 activity was progressively lower at promoter classes that are regulated via sequence-specific transcription factors (ssTFs)—either alone (TFO), in concert with cofactors SAGA, TUP and/or Mediator SWI-SNF (STM), or within an architecture unique to ribosomal protein genes (RP) (Fig. 2B–Ci).

Notably, this pattern of reducing Top2 activity was in direct contrast to relationships between promoter classes and transcriptional activity (measured genome-wide by stranded RNA-seq), which increased across the UNB, TFO, STM and RP promoter classes, respectively (Fig. 2D). We hypothesised that these relationships could be explained by DNA-bound ssTFs sterically hindering access of Top2 to DNA in promoters. Supporting this idea, averaging CC-seq-v2 data around ssTF-bound motifs in the NDR[29] revealed a strong depletion in Top2 activity over a ~30 bp region centred directly at the motif midpoint (Fig. 2E), a feature that was conserved across the binding sites of three common ssTFs expressed throughout the cell cycle (Reb1, Abf1 and Rap1; Fig. 2F), but not at Fkh1, which is degraded in mitosis and therefore absent in these G1-arrested cells (Fig. 2F)[30,31]. Nevertheless, Top2 activity was locally enhanced in the ~100 bp flanking the ssTF-bound regions consistent with these being regions of relative nucleosome depletion, and thus locations in which Top2 can access the DNA substrate more readily.

We next focussed on the quantitative relationship between Top2 activity and transcription. Interestingly, both within promoter classes, and across the total set, we observed a curvilinear relationship, whereby the lowest and highest expressed promoters had the lowest

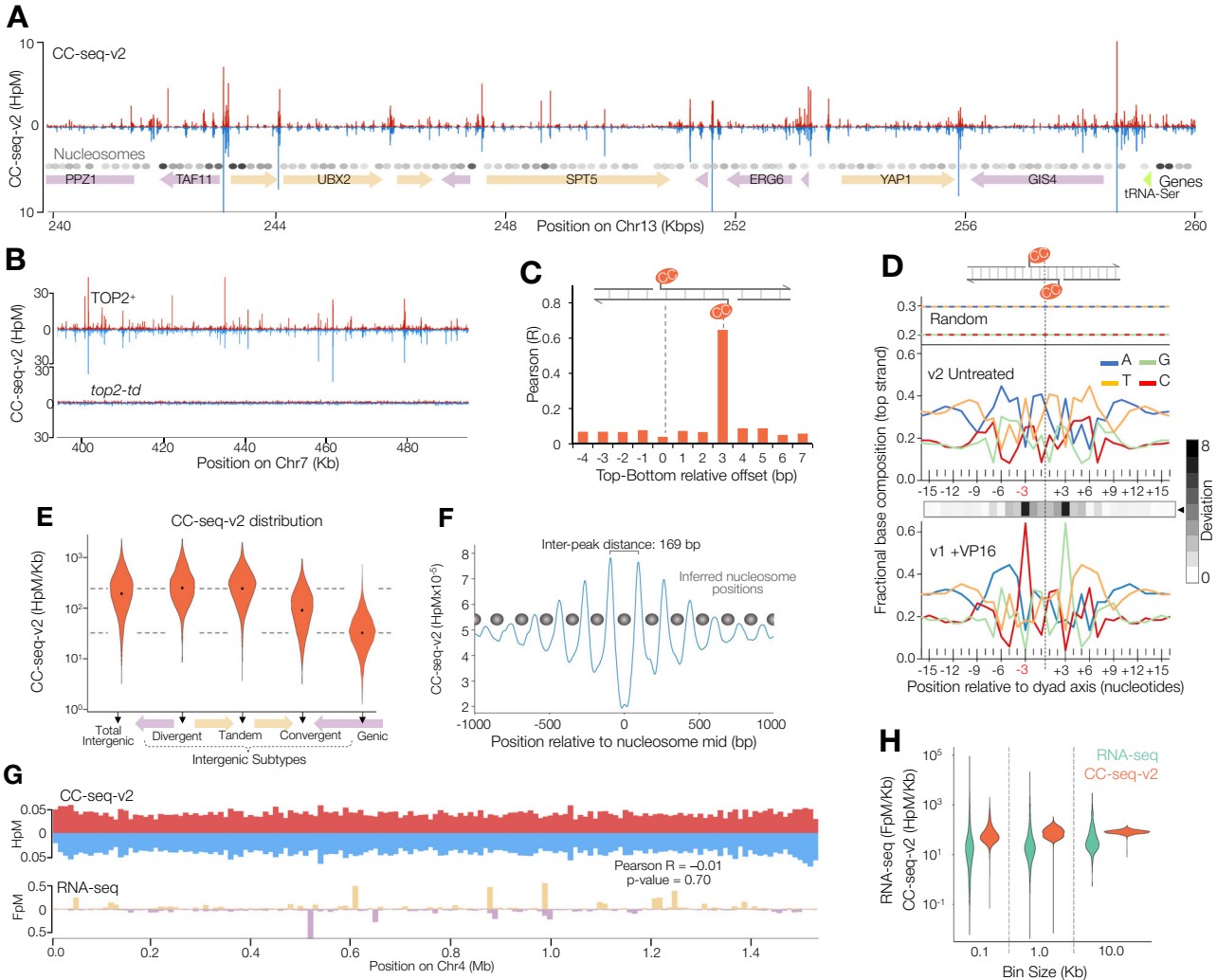

**Fig. 1 | CC-seq-v2 maps physiological Top2 activity in etoposide-unchallenged cells. A** Nucleotide-resolution CC-seq-v2 maps of physiological Top2ccs in G1-arrested *S. cerevisiae* over a 20 kb range. Red and blue traces indicate Top2-linked 5′ DNA termini on the top and bottom strands, respectively. Arrows and rectangles indicate positions of genes and Ty elements, respectively. Circles indicate positions of nucleosome midpoints, with grayscale intensity indicating nucleosome centre positioning score[56]. **B** CC-seq-v2 maps of physiological Top2ccs in *TOP2+* and *top2-td* strains, under degron-inducing conditions (see methods). **C** Pearson correlation (R) of CC-seq-v2 signals on top and bottom strands, after displacement (Top-Bottom) by the indicated distance. Positive values indicate bottom strand displacements leftwards relative to the top strand. **D** Average DNA base composition over a 32 bp window centred on random genomic loci (top), physiological Top2 sites mapped by CC-seq-v2 (middle), or VP16-stabilised Top2cc sites mapped by CC-seq-v1 (bottom)[23]. Values reported are

for the top strand only. Absolute log2 fold nucleotide base deviation between the v2 untreated and v1 VP16 patterns is plotted as a grayscale track, highlighting the greatest difference at the −3 and +3 positions. **E** Violin plot of genome-wide CC-seq-v2 Top2cc signal densities in specific genome compartments. Dotted lines highlight difference between median density in promoter-containing IGRs and genic regions. **F** Average (pileup) of CC-seq-v2 signals around 67548 nucleosome midpoints, smoothed with a 100 bp sliding Hann window. Grey circles indicate inferred positions of nucleosomes. **G** CC-seq-v2 maps of physiological Top2ccs (top) and stranded RNA-seq maps of transcription (bottom) in G1-arrested W303 *S. cerevisiae* cells. Both data sets are binned at 10 kb resolution, and plotted across whole chromosome 4. The *P* value of a Pearson's product-moment correlation two-sided t-test is reported. **H** Violin plot of genome-wide RNA-seq and CC-seq-v2 Top2cc signal densities, binned at three resolutions. Source data are provided as a Source Data file.

Top2 activity, and the medium expressed promoters had the highest (Supplementary Fig. 3A, B). By contrast, Top2 activity within gene bodies was anticorrelated with transcription (Supplementary Fig. 3A, C), whereas Top2 activity within terminators displayed a weak positive relationship with transcription of the upstream gene (Supplementary Fig. 3A, D).

In the compact and gene-dense nature of the *S. cerevisiae* genome, the superhelical state of any locus is likely related to the orientation, length, and activity of all transcription in the local neighbourhood[4]. To simplify this considerable complexity, we focussed on sites of divergent and convergent transcription (Supplementary Fig. 3E–G), in which the superhelical state (negative and positive, respectively)

should be less ambiguous. Notably, Top2 activity was weakly negatively correlated with sum divergent transcription (Supplementary Fig. 3E, G; Pearson $R = −0.17$) and weakly positively correlated with sum convergent transcription activity (Supplementary Fig. 3F, G; Pearson $R = 0.1$). Moreover, directly cross-correlating our stranded RNA-seq maps of transcription with our CC-seq maps of Top2 activity over a displacement range of ±25 Kb revealed that transcription was correlated—albeit again very weakly—both positively and negatively with downstream and upstream Top2 activity, respectively (Supplementary Fig. 3H).

Collectively, these observations suggest Top2 activity on transcription-associated positive supercoils[32] that are predicted to

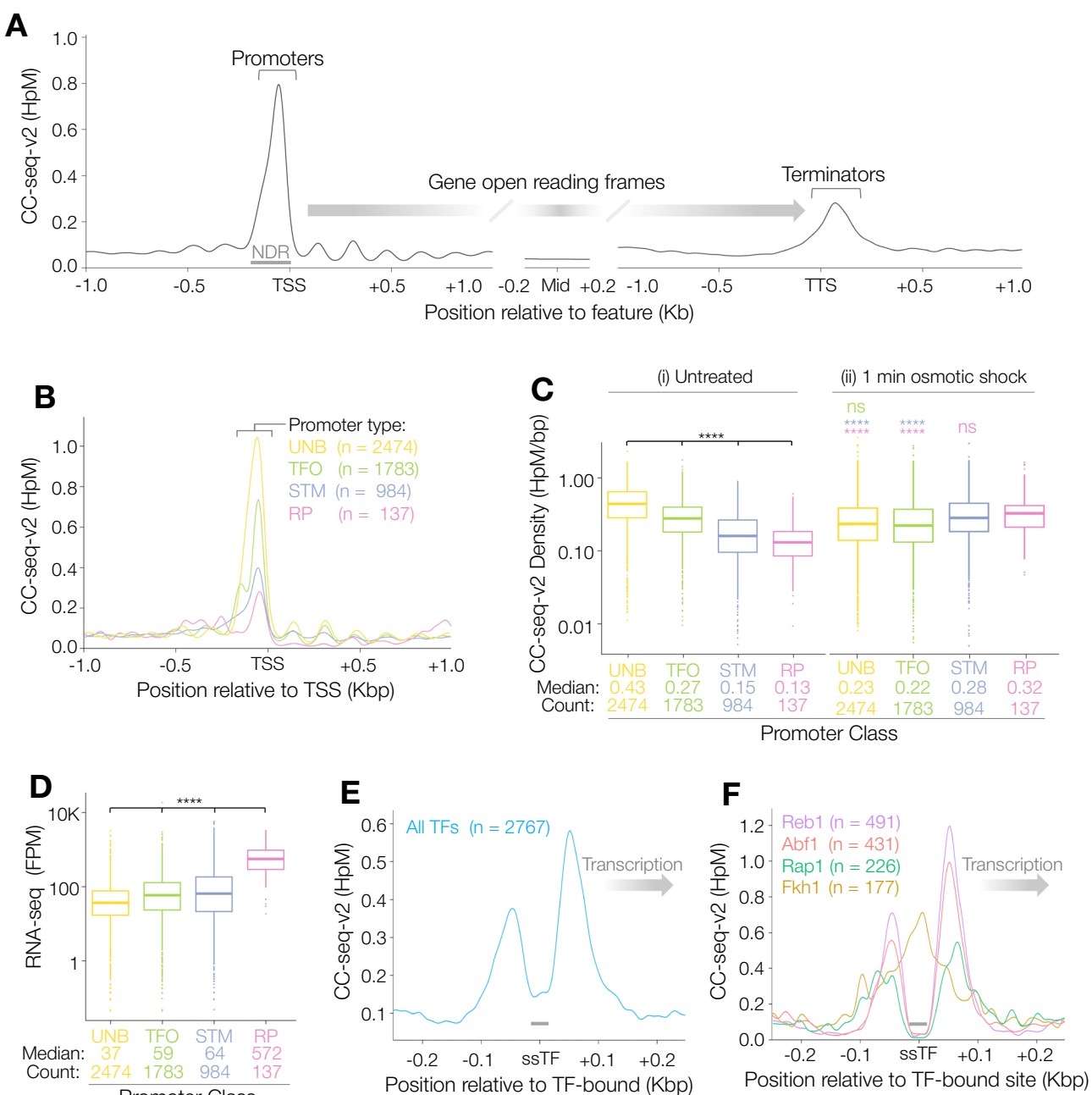

**Fig. 2 | Promoter architecture influences Top2 activity. A** Average (pileup) of CC-seq-v2 Top2cc signals around TSSs (left), gene body midpoints (middle), and TTSs (right). Signals were smoothed with a 100 bp sliding Hann window. **B** Average (pileup) of CC-seq-v2 Top2cc signals around transcription start sites (TSS) within promoters classified previously[28]: those bound by SAGA, TUP and/or Mediator SWI-SNF (STM); general transcription factors only (TFO); unbound by all but the pre-initiation complex (UNB); or those of ribosomal protein genes (RP). Signals were smoothed with a 100 bp sliding Hann window. **C, D** Boxplots of CC-seq-v2 Top2cc signal density in nucleosome depleted/free regions (NDRs) in (i) untreated or (ii) osmotically shocked cells (**C**), or of transcription activity measured by RNA-seq (**D**), for STM, TFO, UNB and RP genes. In all boxplot panels, upper and lower box limit: third and first quartile; bar: median; upper and lower whisker: highest and lowest values within 1.5-fold of the interquartile range. Statistical test used: two-sided K-S. Asterisks indicate Bonferroni-adjusted $P$ values $< 4.19 \times 10^{-4}$, with colours, when present, indicating the specific pairwise comparison being tested. ns, $P \geq 0.838$. **E, F** Average (pileup) of G1-arrested CC-seq-v2 Top2cc signals oriented by local gene transcription direction around all transcription-factor binding sites (**E**), or stratified by the indicated transcription factor classes (**F**). Note of the four classes, Fkh1 is known to be unbound in G1 cells[30,31]. Source data are provided as a Source Data file.

accumulate downstream of promoters and at sites of high convergent transcription[3]. Nevertheless, these local correlations were weak—even at sites of convergent transcription. Thus, neither at local (Supplementary Fig. 3A–H), nor at broad scales (Fig. 1G, H), is physiological Top2 activity well correlated with transcriptional activity—an unexpected finding given the long-standing twin-domain paradigm[1–3].

## Osmotic shock rapidly and transiently remodels chromatin in vivo

In cells, DNA is packaged into chromatin by interactions with a complex ensemble of proteins—including histones, transcription factors, and other DNA interacting proteins. We hypothesised that these interactions—and the higher-order structures they organise—might limit the availability of Top2 substrates at sites of transcription-

induced superhelical stress. Also, it is well established that altering ionic forces can disrupt protein-DNA interactions within chromatin in vitro[25,33,34]. Putting these ideas together, we reasoned that acute ionic shock would perturb protein-DNA interactions in chromatin in vivo, allowing the formation of Top2 substrates at sites of latent topological stress. To test this, we subjected cells to a rapid osmotic shock and assessed the impact on the distribution of Top2 activity relative to sites of protein-DNA interactions within cells.

Validating our ability to perturb chromatin interactions in vivo using this strategy, within seconds of treatment, Top2 activity increased transiently at locations usually occluded by ssTFs (Fig. 3A)—consistent with the known effect of osmotic shock-induced nuclear ion perturbations destabilising ssTF-DNA interactions[35]. Indeed, the overall difference in total Top2 activity at different promoter classes seems to be largely driven by the degree of ssTF-binding, because upon acute osmotic shock these differences were suppressed (compare Fig. 2Ci with Fig. 2Cii).

Prior work has suggested that, in contrast to ssTF-DNA interactions, histone-DNA interactions are more stable throughout acute osmotic shock[35]. Supporting this, whilst a fraction of Top2 activity did redistribute at fine scale from linker DNA to positions usually protected by nucleosomes (Fig. 3B), the fold increase was less than at ssTF-protected sites (Fig. 3B). Despite this small change, closer inspection of the global patterning of Top2 activity around known nucleosome midpoints (Fig. 3C) revealed a transient lengthening of the inter-peak distance from -169 to 176 bp, suggesting an average increase of inter-nucleosomal spacing by 7 bp (Fig. 3D, E) at sites of Top2 activity. Altogether, these observations are consistent with osmotic shock inducing transient structural changes to chromatin in vivo reflected by transient reorganisation of nucleosomes and transcription factors.

Because fine-scale changes in Top2 activity are an indirect readout of chromatin changes, we also directly assessed the impact of osmotic shock on chromatin using MNase-seq. In line with previous observations[35], only minor changes in nucleosome occupancy were detected throughout osmotic shock (Supplementary Fig. 4A, B). Nonetheless, the changes that were observed were spatially correlated with changes in Top2 activity genome wide (Fig. 3F and Supplementary Fig. 4C–E). At the earliest timepoint (0.5 min), changes in nucleosome occupancy were negatively correlated with changes in Top2 activity, before becoming transiently positively correlated (2–3 mins) (Fig. 3F and Fig S4A, B). These observations indicate rapid fluctuations in nucleosome occupancy and/or chromatin organisation induced by osmotic shock that are spatially and temporally linked to changes in Top2 activity. Altogether, this suggests that osmotic shock impacts chromatin at specific sites that are then acted on by Top2.

### Osmotic shock reveals genome-wide correlation of Top2 activity and transcription

We next assessed in more detail the specific sites where Top2 activity increased in response to osmotic chromatin disruption. Within 30 s, Top2 activity redistributed dramatically (Fig. 4A), becoming far more heterogeneous at broad scale (Fig. 4B) due to the formation of hot- and cold-spots across all chromosomes (Fig. 4A, C–E, Supplementary Fig. 5). Notably, this broad-scale heterogeneity was highly transient, with a return of relative homogeneity by 3–5 min (Fig. 4B). To assess this global redistribution of Top2 activity, we cross-correlated each time point of osmotic shock with the untreated control, at fine, medium and broad scales (Fig. 4F). At every scale, correlation dropped dramatically at the earliest time points (0.5–1 min) and recovered over 2–30 min.

If chromatin does indeed obscure topological stress in vivo, we hypothesised that the osmotically induced chromatin perturbations would reveal a stronger association between Top2 activity and transcription. This was indeed the case: genome-wide correlation between RNA-seq maps of transcription and CC-seq maps of Top2 activity increased rapidly (0.5–2 min; Pearson $R = 0.4$), before diminishing

(Fig. 4C–E, G). At finer scales (e.g. 100 bp) the magnitude of these changes was less pronounced, further suggesting that broad-scale phenomena drive these effects (Fig. 4G).

To quantitatively assess the spatial relationship between increased Top2 activity and local transcription in more detail, we repeated the cross-correlation analysis between RNA-seq and CC-seq data, but now over the osmotic shock time course (Fig. 4H). Strikingly, the positive region of the cross-correlation function (CCF), located downstream of transcription (where positive superhelicity is expected to arise), increased rapidly at the peak phase of the shock (0.5–3 min; from Pearson $R = 0.04$ to $R = 0.26$), and broadened considerably at the earliest timepoints (0.5–1 min), extending out to +25 Kb (Fig. 4H). By contrast, changes in the negative region of the CCF, located upstream of transcription (where negative superhelicity is expected to arise), were much weaker.

To probe these effects in more detail, we repeated the analysis of the relationships between Top2 activity and local transcription, but now at the peak of osmotic shock (1 min) (Supplementary Fig. 3I–P). Notably, the curvilinear relationship observed at promoters in unperturbed conditions was substantially flattened (Supplementary Fig. 3I, J), whereas the negative and positive trends observed within gene bodies and termination sites, respectively, were enhanced (Supplementary Fig. 3I, K, L). Strikingly, whereas the negative association was abolished at divergent sites (Supplementary Fig. 3M, O), the positive association was increased at convergent sites (Supplementary Fig. 3N, O). Altogether, these data suggest that normal chromatin organisation conceals transcription-associated positive superhelicity by suppressing the formation of substrates upon which Top2 will act.

### Hotspots of transcription-associated positive superhelical stress

Zooming into the osmotically induced Top2 hotspot regions (Fig. 5A–D, Supplementary Fig. 6A–F) revealed a strong positive association with long genes, especially at regions of convergent transcription (Fig. 5A–C, Supplementary Fig. 6A–E) and/or Ty transposon elements (Fig. 5D, Supplementary Fig. 6F)—all sites of predicted transcription-associated positive superhelicity. To quantify these effects, we averaged Top2 activity around loci of each class. At long genes, Top2 activity was transiently increased in the IGR immediately downstream, and up to 10 kb beyond (Fig. 5E), consistent with osmotic shock revealing positive supercoils over broad distances. Moreover, Top2 activity also increased dramatically throughout regions of convergent transcription (Fig. 5F), where positive supercoils would be expected to accumulate (Supplementary Fig. 3). Top2 activity increased most dramatically at boundaries of Ty elements (Fig. 5G)—known sites of positive superhelicity[36]—where it was highly concentrated within the flanking long terminal repeat (LTR) regions, but also spilled into the Ty coding region itself, which, in unperturbed conditions, was devoid of Top2 activity (Fig. 5G). To directly probe the link between Top2 activity induced by osmotic shock and regions of positive superhelicity, we averaged Top2 activity around previously reported binding sites of ectopically expressed GapR (Fig. 5H)—a known binder of positively supercoiled DNA[36]—revealing a clear association that was not observed at sites of low GapR binding (Supplementary Fig. 6G).

Whilst we favour the view that osmotic shock perturbs chromatin to allow formation of new Top2 substrates, it is formally possible that Top2 activity itself may be impacted by osmotic shock, stabilising Top2ccs in topologically stressed regions. To address this, we directly stabilised Top2 activity with etoposide and checked for a similar phenomenon to osmotic shock. Notably, etoposide treatment failed to reproduce the effect caused by osmotic shock (Supplementary Fig. 6H–J)—and in fact partially reduced the strength of the effect (Supplementary Fig. 6H–J), as would be expected if etoposide-dependent poisoning of pre-existing Top2ccs was inhibiting Top2 activity at newly available substrates formed upon chromatin disruption.

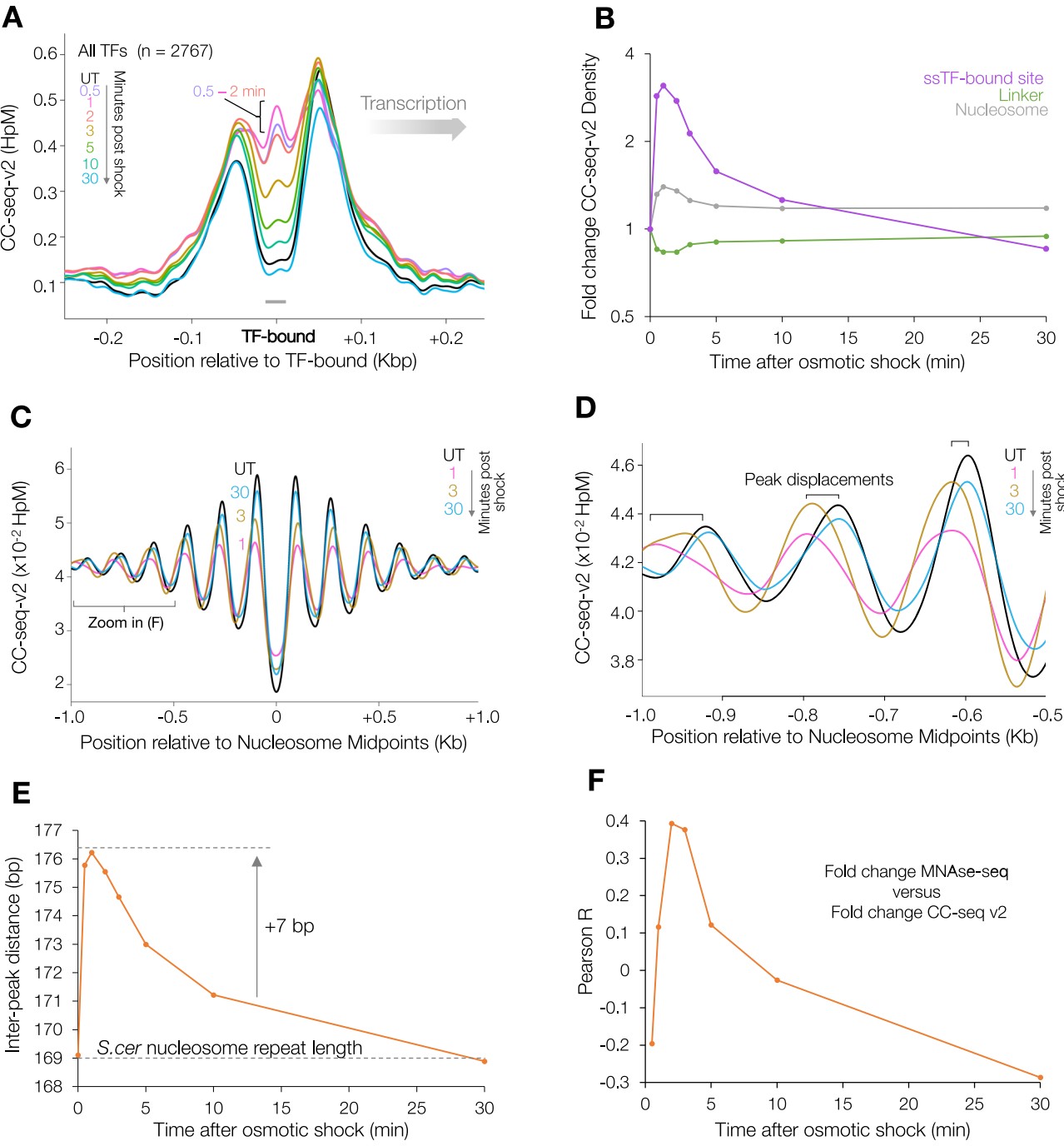

**Fig. 3 | Osmotic shock rapidly and transiently remodels chromatin in vivo.**
**A** Average (pileup) of CC-seq-v2 Top2cc signals oriented by local gene transcription direction around all transcription-factor binding sites in untreated cells and at time points following osmotic shock. **B** Fold change in density of CC-seq-v2 Top2cc signals in 50 bp bins centred on nucleosomal DNA midpoints[56], inferred linker DNA midpoints, or sequence-specific transcription factor binding sites (ssTFs), in untreated cells and at time points following osmotic shock. **C** Average (pileup) of CC-seq-v2 Top2cc signals in a 2 kb region centred on 67548 nucleosome midpoints, in untreated cells and at time points following osmotic shock. Signals were

smoothed with a 100 bp sliding Hann window. **D** Zoom into (C): region −1 to −0.5 kb relative to nucleosome midpoints, showing transient lengthening of the inter-peak distance induced by osmotic shock. **E** The mean interpeak distance (inferred as the inter-nucleosomal distance) in the average (pileup) of CC-seq-v2 Top2cc signals over a 2 kb region centred on 67548 nucleosome midpoints.
**F** Pearson correlation between fold change in MNase-seq and CC-seq-v2 Top2cc signals at each time point following osmotic shock. Source data are provided as a Source Data file.

## Osmotically induced Top2 activity is transcription dependent

Finally, to determine whether the osmotically induced patterning of Top2 activity is directly caused by transcriptional activity, rather than just co-occurring in similar regions due to other aspects of genome organisation, we turned to the *GAL7-10-1* and *GAL2* loci, which can be rapidly transcriptionally induced upon change in the carbon source of

the growth media[37]. Upon induction with galactose for 30, 60 and 90 min we measured a cumulative increase in known galactose-inducible transcripts (Fig. 6A and Supplementary Fig. 7A–B), with the greatest changes (up to ~10,000-fold increases) in *GAL7, GAL10, GAL1* and *GAL2* (Supplementary Fig. 7C–F). Accompanying this acute transcriptional activation, we additionally measured a dramatic change to

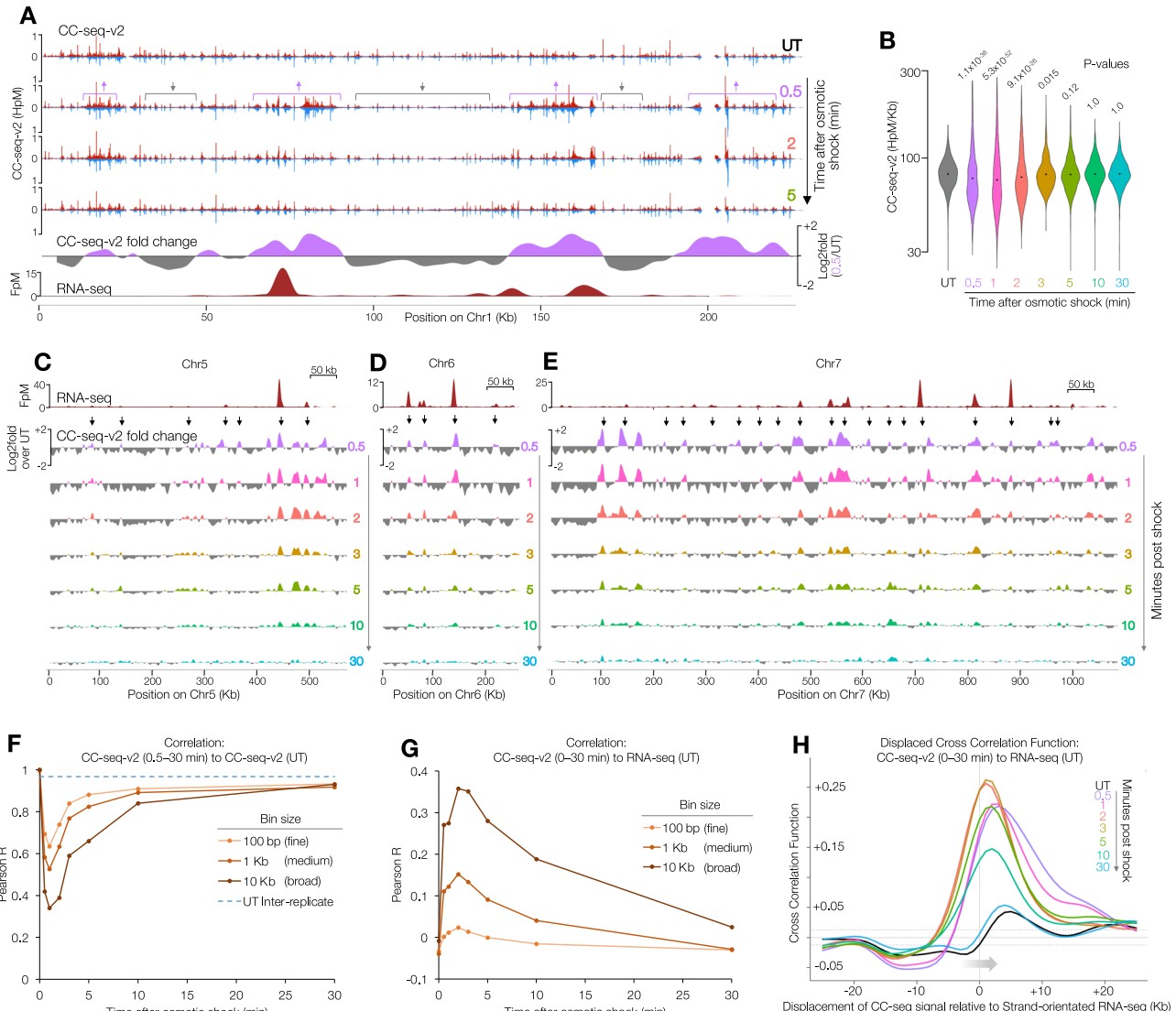

**Fig. 4 | Osmotic shock reveals genome-wide correlation of Top2 activity and transcription. A** 100 bp binned CC-seq-v2 map of Top2ccs across chromosome 1 in untreated cells (UT), and minutes following osmotic shock with 0.6 M sorbitol. Arrows highlight regions of change. The log2 ratio between 0.5 min sorbitol and untreated cells (1 kb smoothed; bottom) reveals regions of relatively greater Top2ccs in each condition overlapping highly expressed regions revealed by RNA-seq smoothed with a 10 kb Hann window (dark red track). **B** Violin plot of CC-seq-v2 density in 10 kb bins, in untreated cells and following osmotic shock. Each timepoint was tested for significant difference vs untreated by two-sided KS test. Bonferroni-adjusted *P* values are indicated above each timepoint. **C–E** Log2 fold change in CC-seq-v2 maps of Top2ccs in minutes following osmotic shock relative to untreated cells (UT) across chromosome 5 (**C**), 6 (**D**), and 7 (**E**), smoothed with a 10 kb Hann window, revealing regions of relatively greater Top2ccs in each comparison and how this changes during the 30 min osmotic shock period (coloured tracks). Black arrows highlight correlation of induced Top2ccs with highly transcribed loci, indicated by RNA-seq smoothed with a 10 kb Hann window (top dark red track). **F** Pearson correlation between CC-seq-v2 signals in untreated cells and in cells shocked for increasing times at three binning resolutions (solid lines). As a control, the inter-replicate Pearson correlation for the untreated condition is also plotted (dotted line). **G** Pearson correlation between CC-seq-v2 and RNA-seq signals at each time point at three resolutions (solid lines). **H** Cross-correlation function (CCF) of 1 kb-binned stranded CC-seq-v2 and RNA-seq signals at displacements of −25 to +25 kb for all timepoints (coloured lines). The arrow indicates the 5′ → 3′ direction of transcription. At positive displacements, RNA-seq signals are positively correlated with 3′-ward CC-seq signals. Dotted lines indicate the 95% confidence interval. Source data are provided as a Source Data file.

local chromosome conformation at *GAL7-10-1* and *GAL2* loci as mea-sured by Hi-C[38] (Fig. 6B)−indicating the formation of strong regions of insulation at these loci extending further than 50 kb. Astonishingly, however, despite these major changes, under osmotically unperturbed conditions we observed negligible increases in Top2 activity at *GAL7-10-1* and *GAL2* (Fig. 6C−E and Supplementary Fig. 8). This remarkable result strongly argues that under normal conditions Top2 activity is not directly correlated with local transcription even in these extreme cases− as might be expected if chromatin organisation impedes the formation of Top2 substrates that are otherwise generated by transcription.

To directly test whether topological stress at these loci was being obscured by chromatin, we subjected transcriptionally induced cells to a two-minute osmotic shock and observed a dramatic, localised−and transcription-dependent−increase in Top2 activity across both the *GAL7-10-1* and *GAL2* loci (Fig. 6C−E and Supplementary Fig. 8). Col-lectively our observations demonstrate that the hotspots of osmoti-cally induced Top2 activity are indeed dependent on local transcription-associated superhelicity, and strongly suggest that that under unperturbed conditions chromatin structure impedes the for-mation of Top2 substrates in vivo.

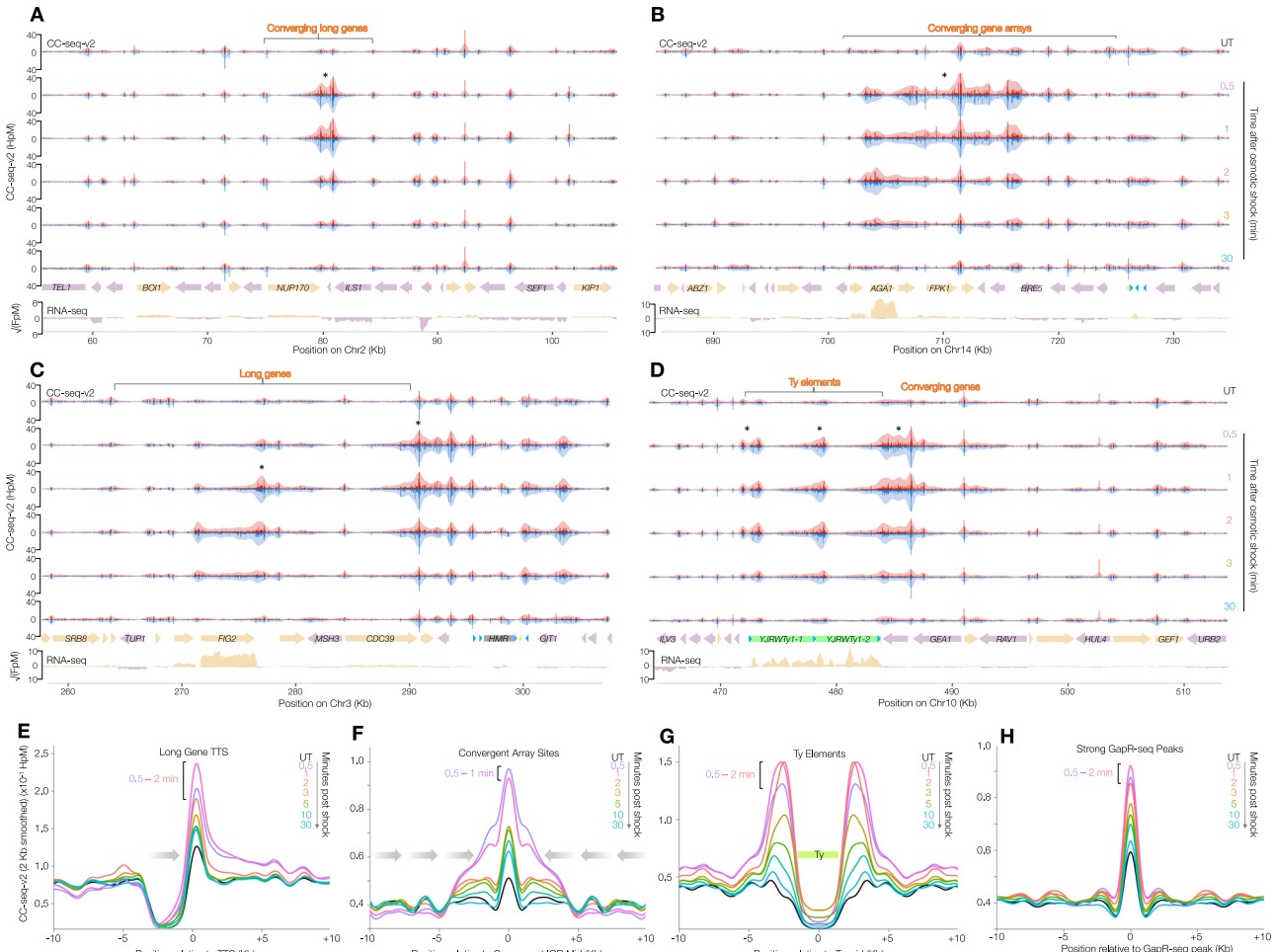

**Fig. 5 | Hotspots of transcription-associated positive superhelical stress.**
**A**–**D** 10 bp binned CC-seq-v2 maps of physiological Top2ccs in G1-arrested *S. cerevisiae*, over 50 kb range at the indicated loci in untreated cells (top) and at the indicated minutes after osmotic shock with 0.6 M sorbitol. Red and blue traces indicate Top2-linked 5′ DNA termini on the top and bottom strands, respectively. In each track, the pale red and blue curves are the same data smoothed with a sliding 1 kb Hann window. Coloured arrows in the bottom panel indicate positions and orientation of specific genomic elements, with stranded RNA-seq data plotted below. **E**–**H)** Average (pileup) of CC-seq-v2 signals around long gene transcription termination sites (TTSs) (**E**), convergent gene arrays (**F**), Ty elements (**G**), and strong GapR-seq peaks (**H**), respectively at the indicated timepoints following osmotic shock. Data are smoothed with a sliding 2 kb Hann window. The peak time points of osmotic shock are indicated by brackets.

## Discussion

The low sensitivity of prior topoisomerase activity mapping methods has generally necessitated the use of catalytic poisons and/or lengthy fixation procedures[23–25,39–41] to probe biological function. Maps generated using these strategies may not be representative of physiological topoisomerase activity, due to biased poisoning of topoisomerases at specific DNA sequences[23], and the consequences of associated DNA damage and of ongoing repair mechanisms. Furthermore, these technical requirements preclude the study of rapid, or transiently changing conditions that might give insight into chromatin dynamics within cells.

Here, by greatly improving the sensitivity of CC-seq, we are able to robustly map transient cleavage complexes that form during the normal catalytic cycle of Top2 within cells. Furthermore, our rapid fixation strategy enables snapshots of physiological Top2 activity with unprecedented sub-minute temporal resolution. These technical breakthroughs will have a broad impact across the topoisomerase field, not least because our antibody-independent method is widely applicable to diverse organisms, including bacteria, yeast, plants and metazoan systems—including mammalian cells and tissues[23] (WHG and MJN, unpublished observations). For example, in mammalian systems, stimuli-induced Top2β promoter activity has been linked to transcription of neuronal early response genes[42] and hormone-response genes[43]. Yet due to technical limitations, it remains unclear whether this Top2β activity is a trigger of, or a response to, induced transcription. With strand-specificity, single nucleotide-resolution, and temporal resolution in the order of seconds, CC-seq-v2 will be able to clarify the role of Top2β within these rapid signal cascades in physiologically relevant, etoposide-free systems.

Probing the relationship between transcription and topological stress within chromosomes has proved challenging due to difficulties in developing methods that directly assess topological change and/or topoisomerase activity itself. Here, we demonstrate that under unperturbed conditions physiological Top2 activity is remarkably uncorrelated with transcriptional activity genome-wide, contrary to expectations based on the longstanding twin-domain model of DNA supercoiling[1–3]. Upon acute hyperosmotic shock, however, Top2 activity is induced at genomic loci predicted to be subject to transcription-induced superhelical stress (long gene TTS, convergent sites, Ty elements, strong GapR peaks). Importantly, this effect—occurring within 30 s—is far more rapid than known transcriptional responses to osmotic shock[44], and was also highly transient, lasting around 5–10 min—in agreement with the known dynamics of nuclear ion concentration changes in response to acute hyperosmotic shock in

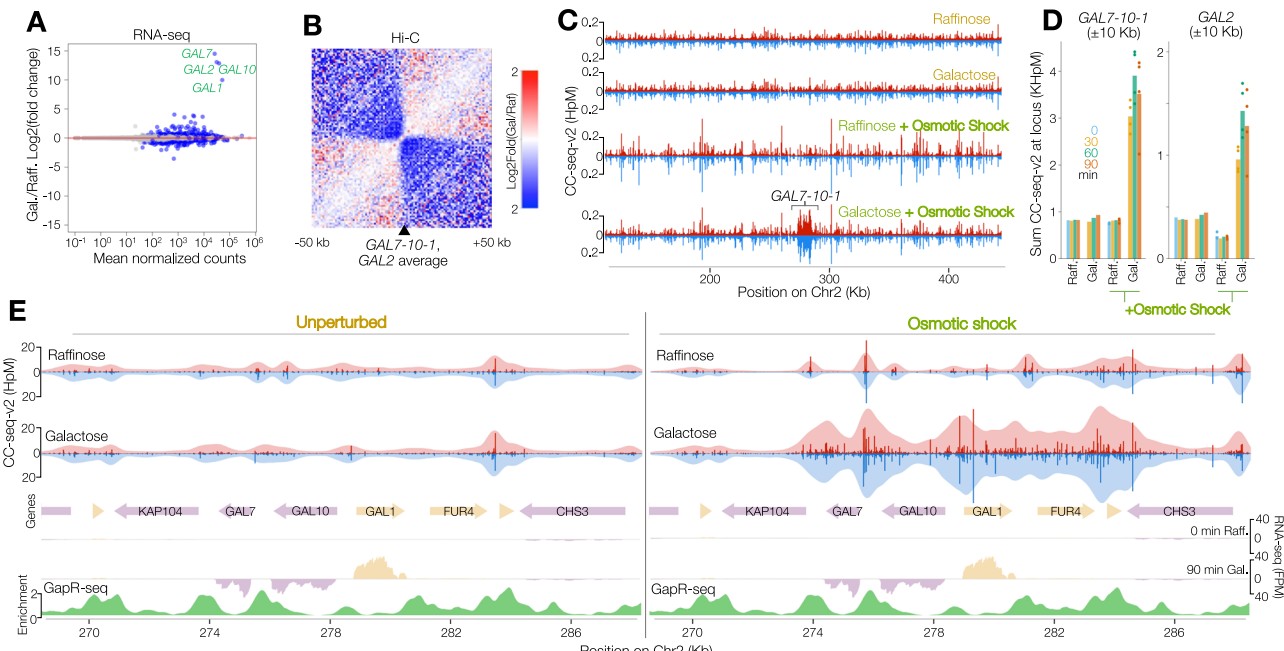

**Fig. 6 | Chromatin buffers transcription-dependent superhelicity. A** MA plot of gene expression changes between cells growing in raffinose (Raff.) media, and those induced with Galactose (Gal.) for 60 min. Grey and blue circles indicate nonsignificant and significant genes (negative binomial GLM two-sided likelihood ratio test FDR-adjusted *P*-value < 0.05), respectively. *GAL1,2,7* and *10* are labelled. **B** Average (pileup) analysis of Hi-C maps of chromosome contact signals at the *GAL7–10–1* and *GAL2* loci, expressed as log2 fold change in contacts in cells growing in galactose for 60 min vs raffinose. **C** 500 bp binned CC-seq-v2 maps of physiological Top2ccs across a 300 kb region centred on the *GAL7–10–1* locus, in raffinose and galactose growth conditions ± osmotic shock for 2 min. In (**C**, **E**) data from individual timepoints (see Fig. 6D and Supplementary Fig. 8A, B) were averaged. **D** Sum CC-seq-v2 signal in a 20 kb region centred on the *GAL7–10–1* locus (left), and *GAL2* locus (right), in cells incubated in raffinose or galactose at 0, 30, 60 and 90 min, with or without subsequent osmotic shock for 2 min. Points indicate independent biological replicates and bars indicate the mean average of these (*n* = 4 for raffinose + OS at 0 min and 90 min; *n* = 4 for galactose + OS at 30 min, 60 min and 90 min), or single replicate values for other conditions. **E** Zoom in to CC-seq-v2 maps of physiological Top2ccs over a 20 kb region centred on the *GAL7-10-1* locus, from cells growing in raffinose (top) vs galactose (bottom), without (left) and with (right) osmotic shock for 2 min. Single-nucleotide resolution signals are plotted as dark red and blue histograms, and a 1 kb-smoothed representation is overlaid in lighter shades. Arrows indicate the orientation of genes (pink and yellow) with stranded RNA-seq data from cells grown in raffinose or induced for 90 min with galactose plotted below them. A GapR-ChIP enrichment map is plotted in green[36]. Source data are provided as a Source Data file.

*S. cerevisiae*[35]. Critically, we have utilised the well-characterised *GAL* gene induction system to demonstrate that induced Top2 activity is directly caused by transcription.

Formally, changes in measured Top2 activity by CC-seq could arise from relative stabilisation of pre-existing Top2ccs, rather than any changes to chromatin that generate new Top2 substrates. However, acute etoposide treatment failed to reveal similar hotspots of Top2 activity, arguing that osmotic shock itself does not increase the half-life of Top2ccs. Moreover, we have observed rapid alterations in chromatin structure that are correlated with changes in Top2 activity, supporting the view that the Top2 activity induced by acute osmotic shock reflects the rapid formation of newly accessible substrates.

Notably, the distribution of induced Top2 hotpots demonstrates that such newly accessible substrates do not form uniformly across the genome but are instead linked to sites of transcription-associated superhelical stress. There are complementary models that could explain this. Osmotic shock could simply destabilise protein-DNA interactions in regions of superhelical stress, generating accessible regions of DNA upon which Top2 opportunistically acts. This is indeed the case at sites of transcription-factor binding and to a lesser extent at nucleosome-bound sites. However, only modest changes in nucleosome occupancy were observed by MNase-seq, and most induced Top2 activity within genes remains strongly patterned by nucleosomes, which would not be expected if Top2 were only acting in regions of nucleosome eviction—suggesting an additional layer of chromatin organisation that has been disrupted.

Single-molecule biophysical and theoretical studies demonstrate that nucleosomal DNA can absorb superhelicity via higher-order

conformational reorganisation[15–22]. However, it has been unclear whether these biomechanical properties are sufficient to buffer superhelicity generated by transcribing RNAP2 in vivo. The poor correlation that we have observed between transcriptional activity and Top2 activity in unperturbed conditions supports the concept that chromatin can absorb superhelical stress in vivo, therein reducing the formation of writhed substrates upon which Top2 might act. Furthermore, based upon the response to acute osmotic shock, we favour the view that induced Top2 activity reflects the rapid formation of writhed substrates at latent sites of topological stress, which arise upon disruption of this intrinsic superhelical buffering capacity of chromatin (Fig. 7). Exploring the specific genetic factors that govern these intrinsic properties of chromatin—and thus how chromatin responds to perturbation—will be of great importance in the future. Moreover, directly characterising the structural conformation of chromatin—and especially its dynamics—will require the further development of advanced tools that can measure topology genome-wide. By utilising the endogenous activity of Top2 as a probe of DNA writhe, our method complements this developing toolkit of assays for mapping genome-wide topology—including bTMP-seq[45] and GapR-seq[36], which preferentially map negative and positive twist, respectively. The great advantages of CC-seq-v2 are that it is minimally invasive, works in situ, and has high spatiotemporal resolution and high dynamic range. Integrating observations from all of these and future methods will help to build a more accurate and comprehensive picture of genome topology.

It is well established that Top2 can act on both positively and negatively supercoiled substrates[8,10] consistent with our observations of

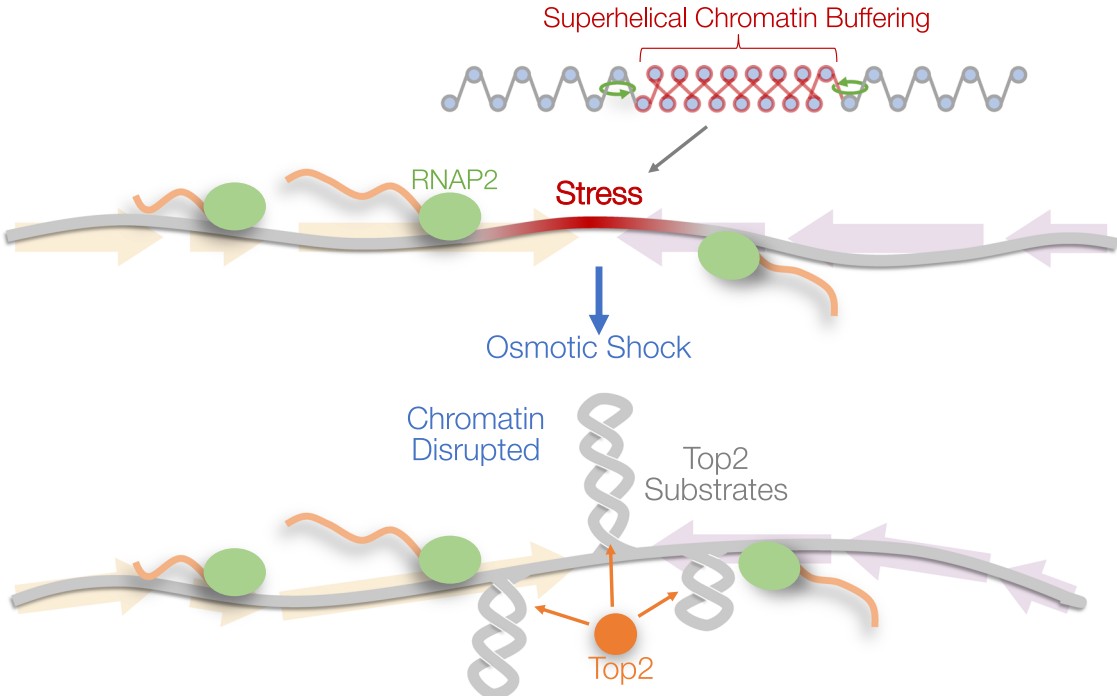

**Fig. 7 | Superhelical chromatin buffering.** Transcribing RNA polymerases (green ovals) generate torsion in DNA that is largely buffered by higher-order structural rearrangements in the chromatin fibre. Acute osmotic shock disrupts this paradigm, via a rapid and transient fluctuation in nuclear ionic forces that alters the structure of the chromatic fibre. Superhelical stress is revealed as plectonemic DNA that is rapidly targeted by Top2 activity.

Top2 activity in both transcriptionally convergent and divergent genomic regions. However, despite this spatial association, we also show that Top2 activity is only positively correlated with convergent transcriptional activity, whereas it is negatively correlated with divergent transcriptional activity. Whilst these observations support a relatively simple model whereby Top2 preferentially acts upon positive supercoils, as has been suggested previously from in vitro studies[32], these relationships are likely to be far more complex in vivo. For example, the anticorrelations between sites of Top2 activity and nucleosome/ssTF binding positions (Fig. 1F and Fig. 2E, F) strongly support the view that features of chromatin organisation other than just local topology influence relative Top2 activity. Indeed, despite the negative correlation between divergent transcription and Top2 activity, the fact that, on average, Top2 activity is higher at divergent IGRs than convergent IGRs is likely to be driven by the relative differences in DNA accessibility at these sites and the fact that positive superhelicity is more able to be absorbed by chromatin than is negative helicity. Complicating matters further, it is also possible that the accessible chromatin state of weakly expressed divergent IGRs become resolution sites for positive supercoils generated at more distal locations, and transmitted through the chromatin over long distances[46,47]. Consistent with this long-range transmission model, we observe that osmotic disruption of chromatin reveals Top2 substrates at convergent points between long arrays of multiple same-sense genes (Fig. 5F). Limiting topoisomerase activity within genes may help to prevent conflict between RNA polymerases and covalent topoisomerase-DNA intermediates arising on supercoils, which might otherwise cause genome instability.

If chromatin can indeed buffer transcription-associated superhelicity, this may question the requirement of Top2 in relaxing supercoils. Much of what is known about topoisomerase function has been gleaned from experiments conducted in osmotically static systems. The sudden ionic effects of osmotic shock upon chromatin may necessitate immediate response by topoisomerases to newly unconstrained supercoils, far in advance of the adaptive transcriptional reprogramming required to promote long-term stress responses.

Supporting this, it is notable that Top2 is able to act throughout acute osmotic shock despite major loss of ssTF-binding[35] (Fig. 3A, C) and changes to the chromatin fibre (Fig. 3C–F).

Intriguingly, supercoiling has been implicated as a regulator of the bacterial osmotic stress response gene *proU*[48]. Notably, selecting for mutations that alter *proU* expression yielded only two affected gene loci: *topA*, encoding bacterial topoisomerase 1, and *osmZ*, encoding H-NS[49], a major member of the nucleoid associated protein (NAP) family that are thought to fulfil similar roles to chromatin proteins found in eukaryotes and archaea[50]. Therefore, the interplay of osmotic stress, genome topology and topoisomerase activity have been known for some time in prokaryotic systems[50,51]. We propose that because ionic fluctuations are likely to proceed even rapid transcription-independent homoeostatic mechanisms[35], the interplay of these phenomena is likely to be ubiquitous across the domains of life. Fascinatingly, Top2 is highly expressed in response to salt stress in plant roots[52], and its ectopic overexpression greatly increases salt tolerance[53]. Whether Top2 is an osmotic/salt stress resistance gene in other organisms remains to be explored.

In summary, we suggest that reorganisation of the chromatin fibre, either via scheduled remodelling associated with normal chromosome metabolism, or via natural environmental fluctuations in osmolarity or other stresses, will necessitate topoisomerase-dependent resolution of supercoiled substrates that would otherwise hinder transcription and DNA replication, and perhaps become sites of genomic instability.

## Methods

### Yeast strains, culture methods and treatment

The *Saccharomyces cerevisiae* yeast strains used in this study are described in Table S1. and were derived using standard genetic techniques. Strains are isogenic to the W303 subtype, mating type a. For all experiments, cells were grown overnight to saturation in 4 mL YPD medium at 30 °C with shaking at 250 rpm, then diluted to OD600 0.5 in 500 mL YPD, or YP-Raffinose (for galactose induction experiments),

and grown until OD600 2.0 in a 2 L baffled flask with antifoam 204 (Sigma A8311) added to 0.0001%. Alpha mating factor was added to a concentration of 20 μg/mL, and the cultures were incubated for a further 90 min. A second dose of alpha factor was added to a final total concentration of 26.6 μg/mL (including first dose), and the cultures were incubated for a further 30 min. At this point >99% of the cells were arrested in G1 (Supplementary Fig. 1D). Cells were either collected at this point (Fig. 1, libraries WG204, WG205, WG206, WG215, WG223, WG239, WG223, WG239), or further treated for specific experiments detailed below.

For the osmotic shock time course (Fig. 2, libraries WG224, WG240, WG225, WG241, WG226, WG242, WG227, WG243, WG228, WG244, WG229, WG245, WG246), a sample of culture was taken as an uninduced control (libraries WG223, WG239), prior to the rapid addition of 6 M sorbitol (pre-warmed to 30 °C) to the media using a 50 mL syringe, to a final concentration of 0.6 M. The culture flask was rapidly mixed on the bench by swirling, and samples were taken at 0.5, 1, 2, 3, 5, 10, and 30 min. The flask was returned to the shaking incubator after the 1 min time point, to maintain temperature at 30 °C.

For the galactose time course (Fig. 3, libraries WG216, WG217, WG218, WG219, WG220, WG221, WG222, WG170, WG171, WG178, WG179, WG199, WG203, WG208, WG209, WG210, WG211, WG172, WG173, WG174, WG175, WG176, WG177, WG200, WG201, WG202, WG212, WG213, WG214), a sample was taken as an uninduced control, before the culture was divided into two and galactose was added to one flask to a final concentration of 2%. Cultures were incubated with shaking at 30 °C, and samples of control and galactose-induced cultures were taken at 30, 60 and 90 min post galactose induction. After every sample was taken (including the uninduced control), a further dose of alpha factor was added to theoretically increase the final concentration by 6.6 μg/mL. This was to counter proteolysis of alpha factor in the media.

For the Top2 degron experiment (Fig. 1B, libraries WG133, WG136), to induce degradation of *top2-td*, galactose and doxycycline were added to cultures of the control and *top2-td* strains (final concentrations of 4% (W/V) and 90 ug/mL, respectively). Cultures were incubated for 60 min, prior to temperature shift to 37 °C for 30 min to induce complete degradation of *top2-td*.

For the etoposide experiment (Supplementary Fig. 6I–K, libraries WG345, WG346, WG347, WG348, WG349, WG350), a sample of untreated *pdr1Δ::DBD-CYC8* culture was incubated, or not, with 100 μM etoposide (VP16) for 15 or 30 min, then subjected, or not, to a 2 min osmotic shock with a final concentration of 0.6 M sorbitol, in a 50 mL falcon tube−prior to immediate fixation with ethanol as described below.

## CC-seq-v2: Cell fixation and collection
CC-seq-v2 is an update of our prior published method that improves sensitivity by several orders of magnitude, enabling the detection of physiological Top2ccs in etoposide-unchallenged cells. Transient physiological Top2ccs were rapidly fixed by denaturation via the direct addition of 70 mL of culture to 150 mL of ethanol (−20 °C) in a 250 mL centrifuge vessel (NALGENE), which had been pre-chilled overnight at −20 °C. Mixtures were immediately vigorously shaken, resulting in a final concentration of 70% ethanol at -0 °C, prior to incubation at −20 °C for 15 min. Vessels were centrifuged at 3500 g for 3 min (4 °C), and the supernatant was aspirated to remove soluble media components that otherwise precipitate during extended incubations at −20 °C. The cell pellet was transferred into 50 mL of fresh 70% ethanol (−20 °C) in 50 mL falcon tubes and incubated at −20˚C for at least 1 h. We have observed that Top2cc CC-seq-v2 signals are fully stable in these preparations for at least 2 weeks. Tubes were centrifuged at 3500 g for 3 min (4 °C) and the supernatant was aspirated fully. Cell pellets were resuspended in 1.2 mL spheroplasting buffer (1 M sorbitol, 50 mM NaHPO4 buffer pH 7.2, 10 mM EDTA) containing 400 μg/mL Zymolyase 100 T (AMS Biotech) and 1% β-mercaptoethanol (Sigma) for 20 min at 37 °C. The entire spheroplast suspension was transferred to

5 mL eppendorfs containing 3.773 ml of ethanol (−20 °C), shaken vigorously to mix, and pelleted by centrifugation at 1000 g for 2 min. In the case of the galactose-induction experiment with osmotic shock and Top2 degron experiment, 70 mL of culture was pelleted at 3500 g for 3 min (RT) prior to resuspension in 1.2 mL spheroplasting buffer (1 M sorbitol, 50 mM NaHPO4 buffer pH 7.2, 10 mM EDTA) containing 400 μg/ml Zymolyase 100T (AMS Biotech) and 1% β-mercaptoethanol (Sigma) for 2 min at 37 °C. The entire spheroplast suspension was transferred to 5 mL eppendorfs containing 3.773 ml of ethanol (−20 °C), shaken vigorously to mix, and pelleted by centrifugation at 1000 g for 2 min. The high sorbitol concentration in the spheroplasting buffer elicits an osmotic shock effect that is highly correlated with the 2 min time point of the osmotic shock time course (Pearson $R = 0.86$).

## CC-seq-v2: Nonproteolyzing phenol-chloroform extraction
The ethanolic supernatant of each sample was fully aspirated prior to addition of 800 μL 1× STE buffer (2% SDS, 0.5 M Tris pH 8.1, 10 mM EDTA, 0.05% bromophenol blue). Cells were lysed using a small plastic pestle (VWR) before addition of a further 1600 μL 1× STE buffer, with mixing by inversion and incubation for 10 min at 65 °C. Samples were cooled on ice and 1250 μL Phenol-Chloroform-isoamyl alcohol (25:24:1; Sigma) was added. The mixtures were emulsified by shaking vigorously 30 times, prior to incubation at room temperature (RT) for 5 min. The mixtures were re-emulsified by shaking vigorously 30 times, prior to phase separation by centrifugation at 20,000 × g for 5 min. We aim to minimise mechanical shearing of the lysate prior to phenol chloroform extraction, to encourage peptides that are covalently linked to high molecular weight DNA to segregate into the aqueous phase alongside other purified nucleic acids. Free protein, coated in SDS, will partition to the interphase. 2 mL of the aqueous phase was removed to a clean microcentrifuge tube, taking care not to disturb the interphase, and nucleic acids were precipitated with 3.5 mL ice-cold ethanol, pelleted by centrifugation, washed with 5 mL ice-cold 70% ethanol, and dissolved in 1 mL 1x TE buffer pH 7.5 overnight at 4 °C.

## CC-seq-v2: Enrichment of covalent protein-linked DNA (CCs)
1 mL aliquots of nonproteolysed DNA (prepared as above) were fully resuspended by heavy vortexing and inversion on a rotation wheel for 15 min (RT), prior to sonication to an average fragment size of 300−400 bp with Covaris (duty cycle: 10%, intensity/peak power incidence: 75 W, cycles/burst: 200, time: 21 min). A sample of 12 uL was removed and incubated at 37 °C for 45 min with 3 μL of RNASE A (50 mg/mL). DNA shearing was checked by 1% agarose gel electrophoresis, and DNA was quantified in technical triplicate with the Qubit (ThermoFisher) and High Sensitivity reagents. Human-Lambda DNA spike (see Internal CC-seq Control, below) was added to 950 uL of sample, then the total volume was made up to 1 mL with 1x TE (pH 7.5). Triton-X100, N-Lauroylsarcosine sodium salt and NaCl were then added to complete the binding buffer (final concentrations: 0.3 M NaCl, 0.2% Triton-X100, 0.1% N-Lauroylsarcosine sodium salt). Each sample was divided over two QIAquick silica fibre membrane spin columns (QIAGEN). Under these conditions, protein, but not nucleic acids, bind to the silica membrane, leading to selective retention of any CCs. The flowthrough was reapplied to the column to improve yield. Columns were washed six times with 600 μL of TEN (10 mM Tris pH 7.5, 1 mM EDTA, 0.3 M NaCl) per 1 min wash to remove any residual non-CC DNA fragments, prior to elution twice with 125 μL TES (10 mM Tris pH 7.5, 1 mM EDTA, 0.5% SDS). The SDS detergent in the elution buffer releases interactions between protein and silica, thereby releasing bound CCs.

## CC-seq-v2: DNA end repair and adapter ligation
Eluted products were pooled to 500 μL in TES and incubated with 1 mg/mL Proteinase K (Sigma) for 60 min at 50 °C, prior to overnight ethanol precipitation at −20 °C with 1.41 mL ethanol, 0.2 mg/mL glycogen and 200 mM NaOAc. The DNA-glycogen precipitate was

pelleted by centrifugation at $20,000 \times g$ for 1 h at 4 °C, washed once with 1.5 mL 70% ethanol without disturbing the pellet, and recentrifuged for 15 min. The supernatant was aspirated, and the pellet was air-dried for 10 min at room temperature, prior to solubilisation in 26 µL 10 mM Tris-HCl pH 7.5. DNA concentration was measured in a 1 µL sample using a Qubit (ThermoFisher) and High Sensitivity reagents. The remaining 25 µL was used as input for one round of end repair and adapter ligation with NEBNext Ultra II DNA Library Preparation kit (NEB), according to manufacturer's instructions, except all steps were conducted at half-scale and a custom P7 adapter was used[23]. It is our understanding that NEBNext Ultra II contains a polymerase capable of blunting ends by either extending the complementary strand towards a ssDNA 5′ extension, or exonucleolytically trimming a ssDNA 3′ extension, but that ssDNA 5′ extensions are not degraded. This kit also provides 3′ dATP terminal transferase activity enabling ligation of adapters containing a terminal 3′ dTTP. The use of custom adapters is to allow differentiation of the sheared end (P7 adapter) from the Top2cc end (P5 adapter). After ligation of the P7 adapter, TE was added to a final volume of 90 uL. DNA was isolated with AMPure XP beads (Beckman Coulter) according to manufacturer's instructions (beads:input of 78:90 µL) and eluted in 85 µL water. Next, 10 uL of 10X Cutsmart buffer and 5 uL Shrimp Alkaline Phosphatase (rSAP; NEB) were added before incubation for 30 min at 37 °C. Dephosphorylation of remaining unligated free DNA ends prevents the possibility of their ligation to the dephospho-P5 (dp-P5) adaptors used in the next round, significantly reducing background noise in the resulting CC-seq maps. DNA was isolated again with AMPure XP beads (Beckman Coulter) according to manufacturer's instructions (beads:input of 83:100 uL) and eluted in 50 µL 10 mM Tris-HCl. Samples were diluted twofold with 50 µL TDP2 reaction buffer (100 mM TrisOAc, 100 mM NaOAc, 2 mM MgOAc, 2 mM DTT, 200 µg/mL BSA) and incubated with 3 µL of 10 µM recombinant zebrafish TDP2 catalytic domain (Addgene plasmid #200512)[54,55] for 1 h at 37 °C. DNA was isolated again with AMPure XP beads (beads:input of 83:103) and eluted in 26 µL 10 mM Tris-HCl. Next, a second round of adapter ligation was conducted using a new custom adapter dp-P5 without prior end repair in order to prevent dephosphorylation of DNA 3′-phosphate termini which may be generated by TDP2 activity on 3′-CCs. After ligation of the dp-P5 adapter to the Top2-cleaved end, sample volume was adjusted to 90 uL with Tris-HCl, DNA was isolated with AMPure XP beads (beads:input of 78:90) and eluted in 16 µL 10 mM Tris-HCl.

### CC-seq-v2: PCR of CC-seq libraries
DNA concentration was measured in 1 µL using a Qubit. The remaining 15 µL was used as template for the NEBNext Ultra II PCR step using universal primer (P5 end) and indexed primers for multiplexing (P7 end), according to manufacturer's instructions (12–15 cycles). PCR reactions were diluted with 50 µL 10 mM Tris-HCl, DNA was isolated with AMPure XP beads (beads:input of 83:100) and eluted in 30 µL 1 mM Tris-HCl pH 8.1, 100 uM EDTA. Sample DNA molarities were quantified using the Bioanalyzer or Tapestation (Agilent).

### CC-seq-v2: Deep sequencing and data analysis of CC-seq libraries
Multiplexed library pools were sequenced on the Illumina MiSeq (Kit v3–150 cycles), Illumina NextSeq 500 (Kit v2–75 cycles), or Illumina NextSeq 1000 (Kit P2–100 cycles), with paired-end read lengths of 75, 42, or 51 bp, respectively. Paired-end reads that passed filter were aligned using bowtie2 (options: -X 1000 –no-discordant –very-sensitive –mp 5,1 –np 0), using MAPQ0 settings, then SAM files processed via terminalMapper (Perl, v5.22.1; https://github.com/Neale-Lab) that computes the coordinates of the protein-linked 5′-terminal nucleotide. For a detailed description of terminalMapper functionality, see our previous CC-seq v1 publication[23]. The reference genome used in this study is our in-house Cer3H4L2 (S. cerevisiae) build, which we generated by the inclusion of the *his4::LEU2* and *leu2::hisG* loci into the Cer3

yeast genome build. Specifically, Cer3H4L2 is identical to the sacCer3 reference genome (R64-1-1), with the addition of two ectopic insertions: 1173 bp of *hisG* sequence inserted at the *LEU2* locus at position 91,965, and 3,077 bp of *LEU2* sequence including 77 bp of associated unidentified bacterial sequence at position 65,684. All subsequent analyses were performed in R (Version 3.5) using RStudio (Version 1.1.383), unless indicated otherwise. Yeast data sets were filtered to exclude the rDNA, the *CUP1-2* locus, the mitochondrial genome and the 2-micron plasmid. The numbers of aligned reads for each library generated and sequenced in this study are detailed in table S2. Final averaged datasets used for generating figures are detailed in table S3.

### CC-seq-v2: Internal CC-seq Control
Sheared etoposide (VP16)-treated human DNA plus sheared phage lambda DNA (NEB) was spiked in following the sonication stage of the protocol to permit internal quality control and potential calibration. This combined spike allows us to (1) estimate true noise based on the lambda DNA proportion, as it should contain no protein-DNA molecules, and to (2) estimate our signal relative to a consistent human Top2cc sample. However, these methods are currently under development and are therefore currently used only for internal quality control. VP16-treated RPE-1 cells were processed as previously described[23] until just after sonication. This was combined with phage lambda DNA (NEB) at a ratio of 1:50 (human DNA:lambda DNA W:W). DNA concentration was quantified by Qubit, and then mixed at a molar ratio of human DNA:yeast DNA of 1:1000. All subsequent stages of the protocol were identical, except read alignment, for which we used both Cer3H4L2 and hg19 builds successively.

### Filtering of CC-seq data
Depending on the analysis—and as indicated in the figure captions and methods—CC-seq data were (1) unfiltered or (2) filtered in the following way. First, single-nucleotide-resolution data on each strand were thresholded to remove peaks less than 0.3 HpM. Subsequently an offset filter was applied, to keep only sites reflecting a stereotypical 4 bp-overhanging Top2cc DSB structure. In our coordinate system, this is achieved by keeping only top strand peaks with a cognate bottom strand peak (also above 0.3 HpM) at the +3 bp position (and vice versa). This filtering step was performed using *CCTools::CCOffset.Filter*.

### Plotting of CC-seq data
Unfiltered fine-scale (nucleotide resolution) maps of Top2 CCs were produced as simple histograms over a specified region using *CCTools::CCMap*. Dark red and blue line heights indicate numbers of 5′-terminal nucleotides detected at that position on the Watson and Crick strands, in units of HpM. Where indicated in the figure caption, smoothed data are also plotted as pale red and blue polygons in addition to the unsmoothed nucleotide-resolution data. This smoothing was applied using a sliding Hann window of the indicated width, as indicated in the figure caption. Broad-scale maps of Top2 CCs were produced using *CCTools::CCBin* by binning nucleotide-resolution data at the resolution indicated in the figure caption, prior to plotting as a histogram.

### Genomic loci of interest used in this study
Coordinates of genome features such were those reported on the Saccharomyces Genome Database (https://www.yeastgenome.org). Promoter classifications (STM, TFO, UNB and RP) and NDR coordinates were as previously reported[28]. Nucleosome midpoint positions and occupancy scores were as previously reported[56]. Sites of strong positive supercoiling (Supplementary Fig. 6F), vs control sites (Supplementary Fig. 6G) were defined as the top 5% of GapR-ChIP enrichment peaks reported previously[36]. Gene transcription activity was measured by stranded RNA-seq, which is a good approximation of ongoing transcriptional activity in *S. cerevisiae*[57]. Long genes were defined as the

10% longest genes after excluding the bottom 20% of expressed genes in order to remove transcriptionally inactive genes from the analysis. Convergent array loci were defined as follows. The genome was segmented into arrays of neighbouring genes that share the same orientation. A convergent site was defined as the midpoint in the intergenic region between two convergent arrays. The transcription activity of all genes in both oppositely oriented arrays was summed, and the top 10% were used.

### Averaging of CC-seq data around loci of interest

Filtered nucleotide-resolution CC-seq data were aggregated within regions of specified size, centred on the loci of interest, using *CCTools::CCPileup*. The resulting sum total HpM was divided by numbers of loci to give a mean HpM per locus.

### Quantification of Top2 activity in defined regions

Filtered nucleotide resolution CC-seq data were tallied within defined regions using *CCTools::CCSum* and are expressed as the distribution of Top2cc signal densities (box-and-whisker plot), and/or plotted vs neighbouring transcriptional activity (sliding window xy plots). For the sliding window analyses, regions were first ordered by associated transcription activity. Moving average Top2cc density was calculated using a Tukey sliding window with a length of 200 points and an $r$ coefficient of 0.25.

### Correlation between top and bottom strand cleavage positions

Unfiltered peak coordinates on the bottom strand were displaced over the range of −20 to +20 bp, relative to top strand coordinates. After each offset, the data was filtered to include only sites with both top and bottom strand hits. The Pearson correlation (r) between the top and bottom strand signal intensities in these n sites was calculated. This analysis was performed using *CCTools::CCOffset*.

### DNA sequence composition around Top2 CCs

Unfiltered CC-seq Top2cc data were first thresholded to remove sites less than 1 HpM. A dyad axis coordinate was defined as the centre point between the Top2-linked nucleotides on the top and bottom strand (that is, the midpoint of the central two base pairs in the four base pair overhang). Top- and bottom-strand DNA sequence, orientated 5′–3′, was averaged ±15 bp around this dyad axis for every site on each strand, using *CCTools::CCSeqbias*. Data are presented as fractional base composition (line plots). To summarise the deviation at each position between the CC-seq-v2 untreated data and the CC-seq-v1 etoposide-treated data, we computed a custom metric. At each position, we summed the absolute logfolds of each base in the CC-seq-v1 + VP16 data over the matching base in the CC-seq-v2 untreated data. We presented this deviation metric as a one-dimensional grayscale heatmap, with white equal to the minimum value (0.17) and black equal to the maximum (8.40).

### RNA-seq library preparation, sequencing and alignment

Total RNA was extracted using the Monarch Total RNA Miniprep Kit, coupled with bead-beating based cell wall disruption using a Fastprep-24 ribolyser (4 × 30 s at 6 m/s velocity, with 1 min on ice slurry between pulses). RNA integrity was checked by TBE-agarose gel electrophoresis, whereupon no significant degradation of the ribosomal RNA bands was observed. Total stranded RNA-seq was performed using the NEBNext Ultra™ II Directional RNA Library Prep Kit for Illumina, coupled with the QIAseq FastSelect −rRNA Yeast Kit, all according to the manufacturer's instructions. cDNA Libraries were quantified by Bioanalyzer (Agilent), prior to sequencing on the Nextseq 500 (Illumina). Data were aligned to the Cer3 (R64-1-1) genome using STAR[58]. Briefly, the genome indexes were first generated using STAR –runMode genomeGenerate, in combination with the Saccharomyces_cerevisiae.R64-

1-1.107.gtf gene coordinate file, and settings --sjdbOverhang 41 and --genomeSAindexNbases 10. Then RNA-seq paired-end FASTQ files were aligned with STAR, with the settings –runThreadN 7 and –alignIntronMax 2500. The numbers of aligned reads for each library generated and sequenced in this study are detailed in table S2. Final averaged datasets used for generating figures are detailed in table S3. Aligned RNA-seq data were processed in several ways, depending on analysis (see below).

### Gene-centric RNA-seq data processing

RNA-seq counts per gene (used for stratifying pileups around specific features, and for sliding window correlation analysis) were outputted directly by STAR. The fourth column, corresponding to reverse-antisense reads (applicable to the directionality of the library prep) was used. Note that data were intentionally expressed as fragments per million (FPM), *without* normalising for gene length as is typically done when processing RNA-seq data for gene expression analysis (e.g to produce units of transcripts per million, TPM; or fragments per kilobase million, FPKM). Ordinarily, this is done to correct for the fact that longer transcripts will fragment into more molecules (and ultimately more reads) in the early stages of library preparation. Thus, dividing by gene length produces a closer measurement of the number of transcripts per cell–and is highly correlated with transcription rate[57]. However, here we are using our RNA-seq data as a proxy to transcription-induced torsional stress, which is predicted to be a product of the transcription rate and the transcript length.

### RNA-seq data preparation for genome-wide plotting and cross correlation with CC-seq

SAM files generated by STAR alignment were sorted and converted to BAM format, using *samtools-sort* and *samtools-view*. The genome-wide coverage on each strand was computed using *deepTools-bamCoverage*, with the options –extendReads and –binSize 10. The resulting Bed-Graph files on each strand were converted to FullMap format, for ease of plotting alongside our CC-seq data within our bespoke *CCTools* R package (https://github.com/WHG1990/CCTools). For in-house compatibility purposes, the FullMap data were converted from Cer3 coordinates to Cer3H4L2 coordinates by increasing the Pos coordinate by the known insert sizes, at the known insert coordinates on chromosome III[59].

### RNA-seq–CC-seq cross correlation analysis

RNA-seq and CC-seq-v2 data were binned genome-wide at 1 Kb resolution using *CCTools::CCBin*, and then smoothed with a 10-bin sliding Hann window. A cross correlation function was then plotted between these two datasets, using the base $R$ function *stats::ccf*, over a lag (displacement) range of -25 to +25 kb. To account for the relationship between RNA-seq strandedness and the orientation of transcription, the analysis was performed (1) between top strand RNA-seq counts and total CC-seq-v2 counts, and (2) between reversed bottom strand RNA-seq counts and reversed total CC-seq-v2 counts. In practice, RNA-seq counts on the top and bottom strands were separated, then the bottom strand counts were reverse orientated and concatenated with the top-strand counts. In the CC-seq data set, the total counts on the top and bottom strand were tandem duplicated. Then a single cross correlation function was performed between the two data sets. This approach ensures accurate correlation values and confidence intervals for the entire top and bottom strand data set, without the need to average two separate cross correlation functions for the top and bottom strand, which may be statistically problematic. In the function call, $x$ was the CC-seq-v2 data and $y$ was the RNA-seq data, such that positive lag values represent a positive displacement (5′ → 3′ orientation) of the CC-seq-v2 data relative to the RNA-seq data, prior to cross correlation.

## RNA-seq differential gene expression analysis

To analyse differential gene transcription upon galactose induction (Fig. 3A and Supplementary Fig. 7), the *DEseq2*[60] R package was employed. Sense-stranded RNA-seq counts output by the STAR alignment were used as input. Four conditions were used (YP-Raffinose and YP-Raffinose + 30, 60, and 90 min Galactose), with two replicates per condition, to build a DEseq2 dataset. To determine which genes exhibited statistically significant changes over the time course, the likelihood ratio test (LRT) was employed using a full model of ~ *replicate + condition* and a reduced model of ~ *replicate*. Effect sizes were shrunk using *DEseq2::lfcShrink* in combination with *apeglm*[61]. Differential gene expression were plotted using a mean-average (MA) plot (*DESeq2::plotMA*), for each timepoint of galactose induction vs YP-Raffinose (Fig. 3A and Supplementary Fig. 7).

## Hi-C library preparation

The protocol presented here has been amended from Belaghzal *et al.*, (2017) for use in budding yeast[38]. For samples assayed by Hi-C, 50 mL (24–40 OD, sufficient for 1 Hi-C library) of W303 haploid cells were fixed with formaldehyde at 3% final concentration at 30 °C, 250 rpm, for 20 min, then quenched by incubating with a final concentration of 0.35 M glycine (2x the volume of formaldehyde added) for an additional 5 min, 30 °C, 250 rpm. Next, cells were washed with dH$_2$O, collected by centrifugation, snap frozen on dry ice and stored at −80 °C prior to library preparation. Fixed cells were thawed on ice, washed in a spheroplasting buffer and digested with 100 µg/mL 100 T Zymolyase in SB containing 1% beta-Mercaptoethanol for 10 min at 35 °C. Cells were washed in restriction enzyme buffer (NEB3.1), chromatin was solubilised by adding SDS to 0.01% and incubating at 65 °C for 5 min. Excess SDS was quenched by addition of Triton X-100 to 1% and chromatin was incubated with 0.92 U/µL of DpnII overnight at 37 °C, 300 rpm. Digestion was terminated by heat inactivation of the restriction enzyme at 65 °C for 20 min prior to DNA ends being filled in with nucleotides, substituting dCTP for biotin-14-dCTP using Klenow fragment DNA polymerase I at 23 °C for 4 h with interval agitation. The sample volume was diluted 2-fold, crosslinked DNA ends ligated at 16 °C for 4 h using 12.67 U/µl of T4 DNA ligase (Invitrogen) in 1X T4 DNA Ligase Buffer (Invitrogen), and crosslinks reversed by overnight incubation at 65 °C in the presence of proteinase K. DNA was purified by phenol/chloroform/isoamyl alcohol extraction and precipitated with ethanol + NaCl, dissolved in TLE and passed through an Amicon 30 kDa column before treating with RNase A at 37 °C for 1 h. Biotin was removed from unligated ends by incubation with T4 DNA polymerase and low abundance of dATP and dGTP (0.03 mM) at 20 °C for 4 h and at 75 °C for 20 min for inactivation of the enzyme. Hi-C libraries were purified using an Amicon 30 kDa column and DNA was subsequently fragmented using a Covaris M220 (Duty factor 20%, 200 cycles/burst, 240 s, 20 °C). Repair of sonicated ends was achieved using T4 DNA polymerase, T4 Polynucleotide Kinase and Klenow fragment DNA polymerase I. Biotinylated fragments were enriched using streptavidin magnetic beads (C1)—for 15 min—and A-tailing, in addition to ligation of NextFlex (Bio Scientific) barcoded adaptors, was performed while the DNA was on beads. Resulting libraries were minimally amplified by PCR and the DNA was isolated with AMPure XP beads (beads:input of 1:1.1) and eluted in 25–40 µl TE prior to quantification of molarity using the Bioanalyser (Agilent).

## Hi-C data processing and analysis

Hi-C sparse matrices were generated at varying spatial resolutions (0.5 kb, 1 kb, 2 kb, 2.5 kb, 5 kb, 10 kb, 25 kb) using the Hi-C Pro pipeline[62] using a W303 reference genome. Subsequent analyses were performed in R (Version 3.6.1) using RStudio (Version 1.1.456), after correcting for read depth differences between Hi-C libraries.

## Aggregation of Hi-C data around *GAL* loci

Hi-C intrachromosomal interaction data at 1 kb resolution was averaged within regions ±50 kb from *S. cerevisiae GAL1–10–7* gene cluster and *GAL2*. Subsetted interaction values were mirrored either side of the loci of interest and the sum, followed by the median of each row (interacting pair of bins) was calculated. A ratio of the median values for each subsetted Hi-C library was taken: ixn.x/ixn.y, and placed on a log2 scale. Each Hi-C library was plotted as a heat map using an arbitrary colour scale with average interaction values mirrored across the diagonal.

## MNase-seq library preparation

MNase-seq was performed as described previously[63], with minor modifications. G1-arrested cell cultures were prepared exactly as for the osmotic shock CC-seq-v2 experiment (see Yeast strains, culture methods, and treatments above). At untreated and treated time points following osmotic shock with 0.6 M sorbitol, cells were fixed by addition of 17.5 mL of culture directly into prepared 50 mL Falcon tubes containing 2.5 mL of 16% paraformaldehyde (final 2% concentration). These were incubated for 15 min at 30 °C in a shaking incubator at 250 rpm, prior to quenching of the formaldehyde by addition of 10 mL 4.3 M TRIS pH 8.0 and subsequent incubation for 5 min at 30 °C. These fixed samples were maintained on ice until collection of all time points, whereupon they were centrifuged at 3500 *g* for 5 min. The supernatant was aspirated, and the fixed cell pellet was resuspended in 1 mL ice-cold TBS and transferred to a microcentrifuge tube. Samples were centrifuged again at 3500 *g* for 5 min, supernatant was aspirated, and the pellets were snap frozen in liquid nitrogen and stored at −80 °C. The following day, pellets were defrosted and spheroplasted with 1 mL spheroplasting buffer (1 M Sorbitol, 50 mM Tris pH 7.5, 5 mM β-mercaptoethanol, 2 mg/mL Zymolyase) at 37 °C for 30 min. Spheroplasts were pulse-pelleted by centrifugation at 20,000 *g* for 15 s and the supernatant was removed. DNA was digested for exactly 10 min with 100 µL digestion buffer (1 M Sorbitol, 50 mM NaCl, 10 mM Tris pH 7.5, 5 mM MgCl2, 1 mM CaCl2, 0.075% NP-40, 0.5 mM Spermidine, 1 mM β-mercaptoethanol) plus 3 µL MNase and 0.3 µL ExoIII (both NEB). Reactions were stopped by the addition of 12.5 µL STOP buffer (50 mM EDTA, 50 mM EGTA) and mixing. RNA and protein were digested by incubation with 3 µL RNAse A (10 mg/mL) at 42 °C for 30 min, followed by incubation with 12.5 µL 10% SDS and 10 µL proteinase K (50 mg/mL) at 65 °C for 1 h. DNA was purified using the Qiagen PCR purification kit, followed by elution in 25 µL 10 mM Tris pH 8.0. DNA digestion was checked by running a 2.5 µL sample on a 2% agarose gel. A clear primary mononucleosomal band was observed, with weaker di and trinucleosomal bands—indicating efficient digestion. DNA 3′-termini were dephosphorylated by the addition of 22.5 µL 2X CutSmart buffer and 1 µL shrimp alkaline phosphatase (NEB) and incubated at 37 °C for 10 min. DNA was re-purified with the Qiagen PCR purification kit, with elution in 25 µL Tris pH 8.0. DNA sequencing libraries were prepared using the NEBNext Ultra II kit (NEB) according to manufacturer's instructions, except at half-scale, with 1 h adaptor ligation times and a 20 min USER incubation. DNA libraries were PCR amplified with dual-end indexing primers (NEB) and 8 reaction cycles, prior to molarity measurement using Qubit (ThermoFisher Scientific) and TapeStation (Agilent). Libraries were pooled at equal molarity and sequenced using the NextSeq 1000 (Illumina). Paired-end reads that passed filter were aligned to Cer3H4L2 using bowtie2 (options: -X 1000 –no-discordant –very-sensitive –mp 5,1 –np 0), using MAPQ0 settings, then SAM files processed via terminalMapper (Perl, v5.22.1; https://github.com/Neale-Lab). Fragment midpoints (reflecting nucleosome midpoints) were computed in R. Two independent experiments were conducted, and the data were averaged.

## Reporting summary

Further information on research design is available in the Nature Portfolio Reporting Summary linked to this article.

## Data availability

Datasets analysed in this manuscript are detailed in Supplementary Table 2 and 3. The CC-seq-v2, RNA-seq and MNase-seq data generated in this study have been deposited in NCBI's Gene Expression Omnibus and are accessible through GEO Series accession number GSE280153, GSE280157, and GSE280155, respectively. The processed CC-seq-v2, RNA-seq and MNase-seq data are also available at GEO. The analysed data generated in this study are provided in the Supplementary Information/Source Data file. Source data are provided with this paper.

## Code availability

Custom code used to analyse data reported in this manuscript are available at the following public GitHub repositories at https://github.com/Neale-Lab/CCTools[64] and https://github.com/Neale-Lab/terminalMapper[65].

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

## Acknowledgements

The authors thank Antony Oliver (University of Sussex) for the gift of recombinant TDP2 catalytic domain, used in our CC-seq library preparation. We thank Jon Baxter (University of Sussex) for discussions and the gift of the *S. cerevisiae* W303 strains used in this study, and Stephen Gray (University of Nottingham) for access to gene deletion strains. W.H.G. was supported by the Biotechnology and Biological Sciences Research Council Discovery Fellowship BB/V005081/1. M.J.N., R.M.A., E.M.W., and G.G.B.B. were supported by the Wellcome Trust Investigator Award 200843/Z/16/Z and Wellcome Trust Discovery Award 225852/Z/22/Z.

## Author contributions

Conceptualisation: W.H.G., M.J.N. Data Curation: W.H.G. Formal Analysis: W.H.G. Funding acquisition: W.H.G., M.J.N. Investigation: W.H.G., R.M.A., E.M.W., M.J.N. Methodology: W.H.G., G.G.B.B., M.J.N. Project administration: W.H.G., M.J.N. Software: W.H.G., M.J.N., G.G.B.B. Supervision: W.H.G., M.J.N. Visualisation: W.H.G., M.J.N. Writing—original draft: W.H.G., M.J.N. Writing—review and editing: W.H.G., M.J.N.

## Competing interests

The authors declare no competing interests.
