## [Transparent Peer Review file · Nature Communications]

Osmotic disruption of chromatin induces Topoisomerase 2 activity at sites of transcriptional stress

Corresponding Author: Professor Matthew Neale

Version 0:

Reviewer comments:

Reviewer #1

(Remarks to the Author)

In this manuscript Gittens et al. refine a previous method to map Top2 activity, which is now sensitive enough to identify ongoing Top2 catalytic cycles without the need to “freeze” the enzyme with poisons. This is a truly important development that will have a strong impact in the field. They then apply this method to map how Top2 activity changes during osmotic stress, concluding that a destabilization of chromatin causes a redistribution of Top2 activity due to a loss its capacity to “buffer” positive DNA supercoiling. I have more doubts about these conclusions, which, although relevant, I do not consider to be sufficiently proven.

First, authors need to show that osmotic stress induces a strong disruption of chromatin in vivo. This is the basis for their experimental model and all the conclusions drawn in the manuscript. As the authors state, all previous evidence for this is in vitro, where conditions are likely to be much harsher. Indeed, they refer to a previously reported stabilization of histone-DNA interactions upon osmotic shock. I don't see how this would fit with a dramatic disruption of chromatin structure. Furthermore, the evidence that they present to demonstrate changes in chromatin structure is only based on measurements of Top2 activity at a fine scale, where only minor changes are observed (an increase of 7bp in inter-nucleosome distance). It is therefore essential to measure the chromatin changes directly, not indirectly with Top2 activity, and strong changes would be required to be able to claim that the supercoiling-buffering capacity of chromatin is lost. Otherwise, the confirmation of a similar effect on Top2 activity upon chromatin disruption with an additional method (maybe chromatin assembly mutants?) would strongly support the case. This may also have some caveats, but presuming a strong disruption of chromatin structure after observing mild changes in inter-nucleosome Top2 activity is a long jump that needs to be experimentally proven.

Second, to validate the model, it is important to show changes in supercoiling upon osmotic stress. It is something that can be done by different genome-wide methods, as the authors discuss, and, in yeast, also directly by using circular minichromosomes. Something difficult to understand to me is, if Top2 activity is a measurement of supercoiling, which is normally buffered by chromatin in the absence of osmotic stress, why does GapR map to these regions in unperturbed conditions? It is therefore essential to demonstrate that osmotic stress affects supercoiling, and not Top2 activity itself. One possibility is that, under osmotic stress, Top1 is globally inhibited, or sequestered to stress-responsive genes, and Top2 needs to take over. In any case, given the functional redundancy between the different topoisomerases, it is an oversimplification to consider Top2 activity as a measurement of DNA supercoiling.

Finally, I do not think that RNA-seq is the best measurement of ongoing transcription to correlate with supercoiling/topoisomerase activity. Any measurement of nascent transcripts would be more valuable. Also, condition-matched datasets would need to be used. In the manuscript Top2 maps at different times after osmotic shock are compared to unperturbed RNA-seq datasets.

Reviewer #2

(Remarks to the Author)

In this manuscript, Gittens et colleagues developed a new version of their previously published CC-seq (named CC-seq v2) that allows for the first time the isolation of physiological Top2CCs without the use of poison-mediated pre-stabilization. They show that physiological Top2 activity can be detected in control cells at promoters and terminators. Notably, Top2 activity correlates poorly with transcription in physiological conditions and is homogenously distributed throughout the

genome. They propose that this is because chromatin can buffer DNA supercoiling, especially positive supercoiling, as shown in in vitro experiments from other groups. They try to disrupt the buffering capacity of chromatin by performing osmotic shock (OS) that induces an immediate (quite striking) redistribution of Top2 activity. In these conditions, Top2cc starts to correlate with transcription and seems to be enriched on regions that are supposed to be positively supercoiled (convergent genes, 3' end of long genes, and Ty elements). Finally, by performing an elegant experiment on the GAL yeast locus, they show that Top2 redistribution following osmotic shock depends on transcription, and they propose this is due to the accumulation of transcription-dependent positive supercoiling, which was previously buffered by high-order chromatin structure.

Overall, I am very enthusiastic about this work. Isolation of Top2cc is challenging, and doing that without using Top2cc stabilizers is an outstanding achievement. Many results are novel and exciting, and the analysis and statistics are robust.

Further improvements of this manuscript are possible, with very few new experiments, some analysis, and some re-wording over strong claims.

Here are my major points:

- The authors should better explain the improvement of CC-seq v2 vs v1. What are the changes introduced, and why did it lead to an improvement? All figures s1A-D are not explained in the text and are left as granted to the reader. The authors should guide the readers more so we can spend less time interpreting technical details in supplementary figures.
- There is no data about a quantitative estimate of how CC-seq v2 performs better than v1. CC-seq-v2 data should show a higher dynamic range than v1, which could be easily visualized with a density plot. Also, an estimate of an improved signal-to-noise ratio could be shown by a Pearson correlation among replicates with progressive reads down sampling for v1 and v2. Could the authors provide these plots, please?
- The term "chromatin disruption" is rather general and vague. What exactly does this mean? In Figure 3, the authors mainly use CC-seq data to show such a disruption. However, all these plots show altered Top2 activity rather than chromatin disruption. It would be nice to see other methods that prove a fundamental change in chromatin structure, like an MNase treatment or an altered DAPI staining. Additionally, the authors show an increase in inter-nucleosomal distance following OS. Does this mean that histone H1 gets evicted from chromatin? This would be better evidence of real high-order chromatin disruption. For these purposes, the authors might also rely on data from the literature.
- The experiment on the GAL locus is very elegant, and the result is very striking. However, it would be nice to see these data complemented with experiments where the authors use transcription inhibitors to kill transcription before OS. Does the Top2 redistribution is hampered genome-wide when transcription is OFF?
- Throughout the manuscript, I get the message that Top2 mainly works on positively supercoiled chromatin, especially after osmotic shock. The fact that Top2 is more active on positive than negative is a fact, as well as that they see enrichment of Top2cc over GAPR positive regions. However, the authors should be more conservative in their claims. These are my reasons:
 1. First of all, and more importantly, there is no direct mapping of DNA supercoiling following osmotic shock in the manuscript. Considering the very short time of OS treatment, both bTMP-seq and GAPR-seq likely do not have the temporal resolution to catch the changes in DNA supercoiling following osmotic shock. Thus, I will not ask for these data, but this limitation remains and should also lead the authors to soften some of the claims in the text.
 2. Top2 can relax both positive and negative supercoiling. For example, the Cortes Ledesma lab has shown that Top2A suppresses the transcription of early genes by removing negative supercoiling at promoters. The authors see quite an enrichment of Top2CC at divergent promoters, which should be negatively supercoiled (sup fig 3E and 2M). This is important because negative supercoiling can be buffered much less than positive by chromatin, which could explain Top2 activity in unperturbed conditions.
 3. Both bTMP-seq and GAPR-seq come with solid limitations. Indeed, positively supercoiled domains inferred by bTMP-seq poorly correlate with GAPR-seq. So, using GAPR-seq as a probe of positive superhelicity should be considered carefully, as there is currently no gold standard technique in mapping DNA supercoiling in vivo.
 4. The same authors have published that in RPE1 cells, Top2CC are enriched in negatively supercoiled domains (defined by bTMP-seq). This is a peculiar difference compared to what they describe here and should be discussed. Does Top2 have a different role in mammals and yeast?
 5. In the GAL induction experiment, the new peaks of Top2 activity arise mainly between unidirectional genes (GAL7 and GAL10) and divergent promoters (GAL10 and GAL1). These two genic configurations are unlikely to generate positive supercoiling.
 6. The authors often refer to plectonemic structures, but no experiments show that these are the structures resolved by Top2.

To be clear, I am not rejecting the model proposed by the authors, but they should state clearly that the data shown here only suggest a potential model. So, sentences like "If chromatin can adequately buffer most transcription-associated superhelicity in *S. cerevisiae*" or "By utilising the endogenous activity of Top2 as a probe of DNA writhe" should be somehow toned down.

Minor points:

- In sup fig 1C is not clear if the rSAP omission (-rSAP) has been done for v1 or v2. If this is done for CC-seq v2, then there is no direct visual comparison between v1 and v2, which would be informative.
- Combined with the Top2 td, the authors should also show a VP16 treated sample (so that both positive and negative

controls are included)

- The authors should explain better Figure 1H, highlighting that the different dynamic ranges for RNA-seq and CC-seq revealed a more homogenous distribution of Top2 vs gene expression.
- The authors should explain the purpose of spiking in lambda DNA in the material and methods section. Is that a negative control different from VP16-treated human DNA that should be lost during the procedure?
- In Figure 2A, there is no need to show the gene expression quartiles as only a single metagenic profile is shown.
- Please use different colors for the time points with OS. These colors are difficult to interpret. Possibly, use a color-blind-friendly palette.
- In sup figure 3, please correct "redistribtion" with "redistribution."
- Supplementary Figure 4 G, H should be moved to a main figure, as it is a significant result. Also, a profile for only control cells around GAPR regions could be shown in Figure 1.

Reviewer #3

(Remarks to the Author)

Gittens et al is a provocative study of native topoisomerase (specifically Top2) cleavage sites in *Saccharomyces cerevisiae*. Unlike previous studies of Top2, Gittens et al develop a novel strategy for trapping and isolating Top2-DNA covalent complexes (Top2-cc) and generating sequencing libraries from Top2-ccs with no amplification bias. Shockingly, Gittens et al find that native Top2 cleavage sites have minimal sequence bias and are uncorrelated with transcription. Next, the authors use an osmotic perturbation (0.6 M sorbitol) to determine that, while global Top2 cleavage sites are unperturbed, new Top2 cleavage sites occur in regions downstream of highly-expressed genes and are associated with positive supercoiling (+sc). The +sc validation occurs via 1) examining correlation with transcription orientation, as +sc accumulates in convergently oriented transcription units, and 2) with GapR-seq, a recently published technology that uses a small DNA-binding protein as a probe for overtwist associated with +sc. Surprisingly, these +sc associated Top2 cleavage sites occur within 1 min and disappear rapidly (<10 minutes). Lastly, the authors show that these transient Top2 cleavage sites cannot be induced by altering transcription alone, as changing transcription by growth in galactose vs raffinose does not impact Top2 cleavage patterns, except upon osmotic treatment.

The data presented by Gittens et al is abundant, well-analyzed, and thought-provoking. That said, the central conclusion - namely, that chromatin buffers +sc and prevents Top2 cleavage of these supercoils - is insufficiently supported due to needed additional studies and by technical considerations (see below). In my opinion, the authors have an interesting study of Top2 activity during steady state and osmotic shock but it is unclear 1) if the osmolarity-induced Top2 cleavage is due to nucleosome rearrangement/buffering or via an alternative mechanism (e.g. slowdown of Top2 religation), and 2) how nucleosome rearrangement and +sc buffering is distinguished. The ms will be significantly strengthened if the authors can show a definitive link between nucleosome rearrangement/buffering and Top2 activity.

Major issues:

1) The authors show that osmotic shock reveals new Top2 cleavage sites that are associated with sites of +sc. The authors posit that osmotic shock perturbs chromatin, but the data to support this model is based on analysis of known TF binding sites or nucleosome positions (Fig. 3). An assay that directly shows chromatin perturbation (e.g. differential nucleosome positioning or DNA accessibility by histone ChIP-seq, MNase sequencing, or DNase I sequencing) at these loci during osmotic shock is necessary to support the core hypothesis. See also point #3 regarding differentiating +sc buffering vs nucleosome repositioning.

An alternative possibility is that osmotic change could impact Top2 catalysis. If sorbitol treatment decreases Top2 reaction rate (e.g. by increased DNA-binding or decreasing DNA religation), that would manifest as topoisomerase "trapping" on a +sc substrate. This could explain the entirety of the results of the ms and could be potentially tested by using a drug that traps Top2 on the DNA (etoposide) and repeating the Top2-cc assay. If Top2 trapping occurs, there should be minimal changes in Top2-cc between the \pm sorbitol conditions. This issue is compounded by the fact that the authors did not benchmark their novel Top2-cc assay against the previous literature (see #3 below). An additional, perhaps unlikely possibility is that there could be Top2 recruiting factor(s) that recognize these regions upon osmotic shock.

2) The Top2-cc profile in this ms differs significantly from previously published yeast etoposide-treated Top2-cc profiles, including a dataset published by the authors themselves. In Gittens et al 2019, etoposide-induced Top2-ccs occurred in divergently transcribed regions and immediately upstream of transcriptional start sites. Why is the native Top2 dataset in this work so different than the previously published etoposide-treated dataset? Is there any reason why Top2 cleavage should have a strong +sc bias given that the enzyme is able to relax both positive and negative supercoils in vitro (Goto and Wang, 1982) and the etoposide-trapped Top2 cleavage sites suggest a bias towards negative supercoiling (Gittens et al 2019)? Is it possible that the "transient" Top2-ccs being captured here are the difficult to resolve Top2-ccs, i.e. peaks indicate Top2 difficulty or "persistent" Top2-ccs? One could then view the height of each Top2-cc peak in this work not as the frequency that a site is cut by Top2, but the religation rate of each individual sequence. Could this interpretation explain the minimal correlation between Top2 cleavage and transcription in the absence of osmotic stress and the fact that Top2 cleavage occurs semi-uniformly in genomes? This could be potentially tested in vitro using purified Top2. It is important to distinguish between these models and to provide a comparison between this and the previous Top2-cc work to understand why these datasets provide such different portraits of Top2 function.

3) The authors posit that +sc is buffered by nucleosomes and utilize the GapR-seq technique to report evidence of +sc in cycling cells. If osmotic shock perturbs chromatin, the authors should distinguish between nucleosome rearrangement and

+sc buffering. If +sc buffering occurs, then the GapR-seq pattern should overlap with the Top2 cleavage pattern during osmotic shock, but in 6E, the GapR-seq patterns are more limited while the Top2 cleavage sites extend significantly past where +sc is captured. This discrepancy should be explained. An alternative possibility is that osmotic shock generally dislodges nucleosomes (independent of +sc) increasing DNA accessibility to Top2. Nucleosome disruption will occur more frequently at transcription termination sites (where +sc accumulates) as terminators are nucleosome-bound while promoters are more nucleosome-free (Chereji et al, 2017).

4) The authors provide no model for the semi-uniform Top2 cleavage patterns observed in unstressed growth. Are the Top2 sites merely the sites that are nucleosome-free? What if anything explains Top2 cleavage patterns? If the model that only persistent Top2 cleavage sites have been captured is correct, the data could be interpreted that in unstressed conditions transcription-associated Top2 activity is too fast to capture, with osmotic shock slowing activity and revealing Top2 association with transcription. This could also explain the results at the galactose locus, in which Top2 activity at GAL is only observed in galactose + osmotic shock.

Minor points:

- 1) Is top2-td the Top2 with the degron tag?
- 2) I found many of the figures difficult to read. The colors and lines are not intense enough to distinguish easily, and the fonts in many cases are too small to be easily legible.

Version 1:

Reviewer comments:

Reviewer #1

(Remarks to the Author)

I appreciate the effort from the authors in addressing the comments in the first review. I am convinced by some of the new results presented, and I acknowledge the technical advance that this study constitutes, as well as the interesting observations presented. However, I still bear some doubts as to whether the proposed model constitutes the most likely explanation for their observations, or whether alternative possibilities (as suggested in the first review) should be considered in more depth. Please note that, although toned down, the main message of the manuscript is still that disrupting the buffering capacity of chromatin results in increased topological stress and increased Top2 activity, so this needs to be proven.

As indicated in the first review, strong changes in chromatin structure would need to be observed to explain a loss in the buffering capacity of chromatin as the cause of the increase in Top2 activity. This has not been observed, as the osmotic shock results in only minor changes in genome-wide MNase sensitivity. It is true that the authors observe a correlation between changes in MNase-seq and CC-seq v2, but this may just reflect the preference of Top2 for nucleosome-free regions, and not, as put forward by the authors, that the changes in chromatin result in increased accumulation of topological stress. The data obtained with the chromatin assembly mutants could support the model proposed, provided that mutants do show a correlation between Top2 activity and transcription. This should be specifically analyzed and included in the manuscript. I do not agree that acute experiments are needed in this regard. If these mutants have stable disrupted chromatin, even if subtle (as transiently observed upon osmotic shock), topological stress and Top2 activity should be increased, if the model proposed by the authors is correct. I consider this essential for the publication of the manuscript.

Regarding the measurement of DNA supercoiling, I agree that it may be challenging to use GapR-seq or bTMP-seq. However, I do not understand why, as suggested in the first review, the authors do not measure supercoiling directly in a yeast mini-chromosome. This would be an unambiguous way to demonstrate that osmotic stress induces supercoiling, and even distinguishing between positive/negative supercoiling and determining the requirement for transcription could be measured relatively easily.

Finally, regarding the correlation with nascent transcription, I do understand the technical difficulty, but at least the authors should have aimed to have condition-matched datasets for correlation. Osmotic stress will change the transcriptional landscape even at those short timepoints, so I do not think it is correct to correlate unperturbed RNA-seq data with CC-seq v2 after osmotic shock. If this is not what authors do, I apologize. In any case, I agree that one would expect, if something, a stronger correlation, so, in contrast to the above points, I do not consider this to be essential.

Reviewer #2

(Remarks to the Author)

The authors have addressed most of my points.

The manuscript has improved, and the claims and statements in the manuscript are now more balanced.

I would have liked to see some genome-wide transcription inhibition experiments, but I see why the authors are reluctant to go in that direction.

I particularly appreciated how the authors have addressed points from the other reviewers with solid experimental work. I recommend acceptance.

Stefano G Manzo

Reviewer #3

(Remarks to the Author)

Gittens et al have resubmitted a revised version of their manuscript describing a new version of Top2 activity mapping showing how activity is sensitive to osmotic state. The novel Top2 mapping technique and the observed rapid changes in Top2 activity are important advances in the field. I commend the authors for the extensive set of experiments in the revised manuscript. The new data and the clear rewrite address all my earlier concerns.

RESPONSE TO REVIEWERS

Summary of changes:

- 1) Presented a benchmarking comparative figure of CC-seq v1 vs CC-seq v2 as requested by Reviewers 2 and 3, (Fig S2, lines 9–103).
- 2) Presented correlated changes to chromatin structure during osmotic shock as requested by all Reviewers (Fig. 5H and S4, lines 196–205).
- 3) Presented evidence that the effects of etoposide treatment do not phenocopy the effects of osmotic shock, as suggested by Reviewer 3 (Fig. S6I–K, lines 260–266).
- 4) Presented supportive pilot data in the rebuttal (below) concerning the phenotypes of chromatin mutants as suggested by Reviewer 1.
- 5) Generally toned down our text throughout the results, including removing “buffering” from the title, in order that we present our model only as a favoured explanation for our observations rather than as definitive proof, as suggested by Reviewer 2. (Title change, lines 54, 168–178, 218–220, 249, 268, 282–283, 289–290, 354, 358).

Point-by-point response to each referee (comments in BLUE, responses in BLACK):

REVIEWER COMMENTS

Reviewer #1 (Remarks to the Author):

In this manuscript Gittens et al. refine a previous method to map Top2 activity, which is now sensitive enough to identify ongoing Top2 catalytic cycles without the need to “freeze” the enzyme with poisons. This is a truly important development that will have a strong impact in the field. They then apply this method to map how Top2 activity changes during osmotic stress, concluding that a destabilization of chromatin causes a redistribution of Top2 activity due to a loss its capacity to “buffer” positive DNA supercoiling. I have more doubts about these conclusions, which, although relevant, I do not consider to be sufficiently proven.

First, authors need to show that osmotic stress induces a strong disruption of chromatin in vivo. This is the basis for their experimental model and all the conclusions drawn in the manuscript. As the authors state, all previous evidence for this is in vitro, where conditions are likely to be much harsher. Indeed, they refer to a previously reported stabilization of histone-DNA interactions upon osmotic shock. I don't see how this would fit with a dramatic disruption of chromatin structure. Furthermore, the evidence that they present to demonstrate changes in chromatin structure is only based on measurements of Top2 activity at a fine scale, where only minor changes are observed (an increase of 7bp in inter-nucleosome distance). It is therefore essential to measure the chromatin changes directly, not indirectly with Top2 activity, and strong changes would be required to be able to claim that the supercoiling-buffering capacity of chromatin is lost. Otherwise, the confirmation of a similar effect on Top2 activity upon chromatin disruption with an additional method (maybe chromatin assembly mutants?) would strongly support the case. This may also have some caveats, but presuming a strong disruption of chromatin structure after observing mild changes in inter-nucleosome Top2 activity is a long jump that needs to be experimentally proven.

Second, to validate the model, it is important to show changes in supercoiling upon osmotic stress. It is something that can be done by different genome-wide methods, as the authors discuss, and, in yeast, also directly by using circular minichromosomes. Something difficult to understand to me is, if Top2 activity is a measurement of supercoiling, which is normally buffered by chromatin in the absence of osmotic stress, why does GapR map to these regions in unperturbed conditions? It is therefore essential to demonstrate that osmotic stress affects supercoiling, and not Top2 activity itself. One possibility is that, under osmotic stress, Top1 is globally inhibited, or sequestered to stress-responsive genes, and Top2 needs to take over. In any case, given the functional redundancy between the different topoisomerases, it is an oversimplification to consider Top2 activity as a measurement of DNA supercoiling.

We thank the reviewer for their detailed comments and suggestions. As described in the summary response, we have performed MNase-seq that demonstrate changes to chromatin arising upon osmotic shock are spatially correlated with regions where Top2 activity increases, suggesting a mechanistic connection (Fig. 5H and S4, lines 196–205). Due to the timescales over which the phenomena we describe occur (1-2 minutes), which has never been investigated before, neither GapR-seq nor bTMP-seq are suitable probes of changes to chromatin topology because both require long inductions and/or incubations, and may themselves indirectly alter chromatin state. The fact that GapR-seq positively correlates with our dataset should be seen as a positive, but it should also be recognised that such GapR datasets are collected after prolonged GapR expression, which may enable it to slowly accumulate in such topologically stressed regions over time.

We agree that the relationship between sites of Top2 activity and supercoiling is likely to be complex and, whilst we favour our model, we have also toned down our language in a number of areas to take into account this degree of uncertainty (Title change, lines 54, 168–178, 218–220, 249, 268, 282–283, 289–290, 354, 358).

Thirdly, we have also attempted to test our model by a different approach, using chromatin mutants as suggested by the reviewer. For this, we conducted a pilot screen to check if Top2 activity in any of these were phenocopied by the osmotic shock (data included below). Modifying nucleosome organisation by loss of Chd1 (PMIDs: 26175451, 21940898), or lowering global histone levels by loss of the histone transcription factors Spt10 or Spt21 (PMIDs: 8035801 23450643 25228766), both stimulated Top2 activity in 3' gene bodies, long gene TTSs and convergent loci—similar to the effect of acute osmotic shock. However we feel strongly that these data are beyond the scope of this paper, because they contain caveats (as noted by Reviewer 1) that would require considerable additional experimentation. For example, we would need to generate ways of rapidly degrading/depleting chromatin assembly factors because distributions of transcription and topological stress (and therefore Top2 activity) in the constitutive mutants is likely to be affected at steady state. Such experiments could take months-to-years to set up and analyse, and are therefore very clearly follow up work that is beyond the scope of the current study, which is already extensive, now encompassing 84 genome-wide datasets (22 were added in this revision).

Chromatin assembly mutants phenocopy osmotic shock-induced changes in Top2 activity. (A–B) Average (pileup) of CC-seq-v2 signals around long gene transcription termination sites (TTSs) and convergent gene arrays at the indicated timepoints following osmotic shock. Data are smoothed with a sliding 2 kb Hann window.

Finally, I do not think that RNA-seq is the best measurement of ongoing transcription to correlate with supercoiling/topoisomerase activity. Any measurement of nascent transcripts would be more valuable. Also, condition-matched datasets would need to be used. In the manuscript Top2 maps at different times after osmotic shock are compared to unperturbed RNA-seq datasets.

Whilst nascent transcript analysis may be a more direct indicator of topological stress, our data clearly demonstrate a positive relationship between Top2 activity and steady-state RNA transcript levels, so such (challenging) experiments are not necessary to support the point we make. Using nascent transcription is expected to only make the already detected correlations stronger. Moreover, it is technically infeasible to measure changes in nascent transcription rate over the sub-minute timescales of our study with current methodology. Even the shortest transcription pulses are in the order of 10 minutes, ten times longer than it takes for Top2 activity to peak and already begin to decline.

Reviewer #2 (Remarks to the Author):

In this manuscript, Gittens et colleagues developed a new version of their previously published CC-seq (named CC-seq v2) that allows for the first time the isolation of physiological Top2CCs without the use of poison-mediated pre-stabilization. They show that physiological Top2 activity can be detected in control cells at promoters and terminators. Notably, Top2 activity correlates poorly with transcription in physiological conditions and is homogeneously distributed throughout the genome. They propose that this is because chromatin can buffer DNA supercoiling, especially positive supercoiling, as shown in in vitro experiments from other groups. They try to disrupt the buffering capacity of chromatin by performing osmotic shock (OS) that induces an immediate (quite striking) redistribution of Top2 activity. In these conditions, Top2cc starts to correlate with transcription and seems to be enriched on regions that are supposed to be positively supercoiled (convergent genes, 3'end of long genes, and Ty elements). Finally, by

performing an elegant experiment on the GAL yeast locus, they show that Top2 redistribution following osmotic shock depends on transcription, and they propose this is due to the accumulation of transcription-dependent positive supercoiling, which was previously buffered by high-order chromatin structure.

Overall, I am very enthusiastic about this work. Isolation of Top2cc is challenging, and doing that without using Top2cc stabilizers is an outstanding achievement. Many results are novel and exciting, and the analysis and statistics are robust.

We thank the reviewer for recognising the significant technical advances that are presented in our work.

Further improvements of this manuscript are possible, with very few new experiments, some analysis, and some re-wording over strong claims.

Here are my major points:

- The authors should better explain the improvement of CC-seq v2 vs v1. What are the changes introduced, and why did it lead to an improvement? All figures s1A-D are not explained in the text and are left as granted to the reader. The authors should guide the readers more so we can spend less time interpreting technical details in supplementary figures.

- There is no data about a quantitative estimate of how CC-seq v2 performs better than v1. CC-seq-v2 data should show a higher dynamic range than v1, which could be easily visualized with a density plot. Also, an estimate of an improved signal-to-noise ratio could be shown by a Pearson correlation among replicates with progressive reads down sampling for v1 and v2. Could the authors provide these plots, please?

Many thanks for these suggestions. Previously we had confined most technical descriptions of the v2 improvements to our Materials and Methods section. However, we agree that the technical breakthroughs of v2 shouldn't be understated, so we have now (1) highlighted key improvements in the main text (lines 72–77), (2) linked these to the individual Fig S1 panels more clearly, and (3) as requested, included a new supplementary figure that compares data generated by CC-seq-v1 and CC-seq-v2 (Fig. S2, lines 9–103). Please note the requested density plot (Fig. S2H), with inset tabulated standard deviations, and a stronger finescale top/bottom strand correlation when offset by 3 bp (Fig. S2IH) as would be expected if CC-seq-v2 has higher signal to noise than even the etoposide-treated v1 maps.

- The term “chromatin disruption” is rather general and vague. What exactly does this mean? In Figure 3, the authors mainly use CC-seq data to show such a disruption. However, all these plots show altered Top2 activity rather than chromatin disruption. It would be nice to see other methods that prove a fundamental change in chromatin structure, like an MNase treatment or an altered DAPI staining. Additionally, the authors show an increase in inter-nucleosomal distance following OS. Does this mean that histone H1 gets evicted from chromatin? This would be better evidence of real high-order chromatin disruption. For these purposes, the authors might also rely on data from the literature.

We thank the reviewer for these helpful comments, which are in-line with those of reviewers 1 and 3. We have now performed MNase-seq experiments throughout our osmotic shock time course, in duplicate (Fig. 5H and S4, lines 196–205). These show subtle changes in nucleosome occupancy that are spatially and temporally correlated with the more dramatic redistribution of Top2 activity, lending support to our model that osmotic shock induces chromatin perturbation that causes increased formation of Top2 substrates at sites of transcriptional stress. Despite this, we acknowledge that defining the details of our favoured model regarding chromatin buffering of topological stress is beyond the technical capacity of current tools. Therefore we have toned down definitive claims about chromatin buffering of topology throughout our manuscript, and clarified that it is our favoured model to explain our observations (Title change, lines 54, 168–178, 218–220, 249, 268, 282–283, 289–290, 354, 358).

- The experiment on the GAL locus is very elegant, and the result is very striking. However, it would be nice to see these data complemented with experiments where the authors use transcription inhibitors to kill transcription before OS. Does the Top2 redistribution is hampered genome-wide when transcription is OFF?

Whilst we agree that global transcription inhibition might complement this work, we feel that the link between transcription and osmotic shock-induced Top2 activity is already well supported in our manuscript—both via correlation with RNA-seq maps of transcription activity, and via the Galactose induction experiment noted by the referee. Furthermore, global transcription inhibition is not trivial in *S. cerevisiae*, as many compounds are poorly uptaken, non-potent or unspecific (PMID: 21922053), and/or with indirect/uncharacterized modes of action (PMID: 38214235). Furthermore, many global transcription inhibitors induce knock-on effects that could complicate analysis. In the interest of limited time and resources, we therefore focussed our experimental efforts on addressing other aspects raised by the reviewers.

- Throughout the manuscript, I get the message that Top2 mainly works on positively supercoiled chromatin, especially after osmotic shock. The fact that Top2 is more active on positive than negative is a fact, as well as that they see enrichment of Top2cc over GAPR positive regions. However, the authors should be more conservative in their claims. These are my reasons:

1. First of all, and more importantly, there is no direct mapping of DNA supercoiling following osmotic shock in the manuscript. Considering the very short time of OS treatment, both bTMP-seq and GAPr-seq likely do not have the temporal resolution to catch the changes in DNA supercoiling following osmotic shock. Thus, I will not ask for these data, but this limitation remains and should also lead the authors to soften some of the claims in the text.

2. Top2 can relax both positive and negative supercoiling. For example, the Cortes Ledesma lab has shown that Top2A suppresses the transcription of early genes by removing negative supercoiling at promoters. The authors see quite an enrichment of Top2CC at divergent promoters, which should be negatively supercoiled (sup fig 3E and 2M). This is important because negative supercoiling can be buffered much less than positive by chromatin, which could explain Top2 activity in unperturbed conditions.

3. Both bTMP-seq and GAPR-seq come with solid limitations. Indeed, positively supercoiled domains inferred by bTMP-seq poorly correlate with GAPR-seq. So, using GAPR-seq as a probe of positive superhelicity should be considered carefully, as there is currently no gold standard technique in mapping DNA supercoiling *in vivo*.
4. The same authors have published that in RPE1 cells, Top2CC are enriched in negatively supercoiled domains (defined by bTMP-seq). This is a peculiar difference compared to what they describe here and should be discussed. Does Top2 have a different role in mammals and yeast?
5. In the GAL induction experiment, the new peaks of Top2 activity arise mainly between unidirectional genes (GAL7 and GAL10) and divergent promoters (GAL10 and GAL1). These two genic configurations are unlikely to generate positive supercoiling.
6. The authors often refer to plectonemic structures, but no experiments show that these are the structures resolved by Top2.

To be clear, I am not rejecting the model proposed by the authors, but they should state clearly that the data shown here only suggest a potential model. So, sentences like “If chromatin can adequately buffer most transcription-associated superhelicity in *S. cerevisiae*” or “By utilising the endogenous activity of Top2 as a probe of DNA writhe” should be somehow toned down.

We are grateful for these astute points, and the overall message urging caution with regards to Top2 preference for positive vs negative supercoiling. We agree that there are currently no perfect direct methods to map either twist or writhe *in vivo*, and that all such methods are likely subject to biases caused by chromatin accessibility etc. As the reviewer notes, this is exemplified by the fact that current gold-standard genomic methods to map positive and negative twist (bTMP-seq and GAPR-seq) are actually moderately positively correlated across the yeast genome (Pearson R = 0.32). To accommodate these points, we have made changes to the manuscript text to tone down definitive claims about Top2 preference for positive vs negative supercoils (lines 160, 251, and 342-361).

Minor points:

- In sup fig 1C is not clear if the rSAP omission (-rSAP) has been done for v1 or v2. If this is done for CC-seq v2, then there is no direct visual comparison between v1 and v2, which would be informative.
- Combined with the Top2 td, the authors should also show a VP16 treated sample (so that both positive and negative controls are included)

To address these points, we have clarified in the legend that the all tracks are CC-seq “v2” (with the upper being a modified version -rSAP). Further, we have now included a new figure (Fig. S2, lines 9–103) that directly compared (1) Untreated CC-seq-v1, (2) VP16-treated CC-seq-v1, and (3) Untreated CC-seq-v2 data. Please see our response to Referee 3 for further details.

- The authors should explain better Figure 1H, highlighting that the different dynamic ranges for RNA-seq and CC-seq revealed a more homogenous distribution of Top2 vs gene expression.

We have clarified this point in the text (lines 110–111).

- The authors should explain the purpose of spiking in lambda DNA in the material and methods section. Is that a negative control different from VP16-treated human DNA that should be lost during the procedure?

We have expanded our description of the internal DNA spike procedure in the methods, as suggested (lines 703–706). In short, the referee is correct that the lambda DNA should not be bound by the column, nor processed correctly into a sequenceable molecule by our library prep. Therefore, it provides a metric of “true noise” in the libraries, which we use internally for QC purposes. Importantly, this spike is not used in any downstream processing of the data in this study.

- In Figure 2A, there is no need to show the gene expression quartiles as only a single metagenic profile is shown.

Thanks for spotting this. We have corrected the figure.

- Please use different colors for the time points with OS. These colors are difficult to interpret. Possibly, use a color-blind-friendly palette.

Thanks for this feedback, in line with comments from Referee 3. We have now changed this colour palette throughout the manuscript to improve visual identification. Our initial intention was to communicate the directionality of the effect, whilst still allowing distinction between specific timepoints. However this is a tricky (and somewhat mutually exclusive) balancing act, and we acknowledge that certain colours were not easily distinguished, and that it was not helpful to have the UT and 30 min recovery time point labelled in a similar blue colour. Therefore, for the untreated sample we have replaced blue with black, then subdivided the colour wheel equally for the remaining seven osmotic shock time points and increased the line width, which improves visual distinction. Finally, to avoid any remaining ambiguity, we have added inset text indicators to highlight the peak time points of osmotic shock (e.g. 0.5 to 2 min).

- In sup figure 3, please correct “redistirbution” with” redistribution.”

This typo has been corrected.

- Supplementary Figure 4 G, H should be moved to a main figure, as it is a significant result. Also, a profile for only control cells around GAPR regions could be shown in Figure 1.

Despite the caveats identified above regarding GapR-seq, we agree that this comparison is an important point and have moved the old Supplementary Figure 4G to the new main Figure 5H. In the interest of space, rather than reproduce these data and add another panel in Fig 1, the untreated profile referred to is the black line in Fig 5H.

Reviewer #3 (Remarks to the Author):

Gittens et al is a provocative study of native topoisomerase (specifically Top2) cleavage sites in *Saccharomyces cerevisiae*. Unlike previous studies of Top2, Gittens et al develop a novel strategy for trapping and isolating Top2-DNA covalent complexes (Top2-cc) and generating sequencing libraries from Top2-ccs with no amplification bias. Shockingly, Gittens et al find that native Top2 cleavage sites have minimal sequence bias and are uncorrelated with transcription. Next, the authors use an osmotic perturbation (0.6 M sorbitol) to determine that, while global Top2 cleavage sites are unperturbed, new Top2 cleavage sites occur in regions downstream of highly-expressed genes and are associated with positive supercoiling (+sc). The +sc validation occurs via 1) examining correlation with transcription orientation, as +sc accumulates in convergently oriented transcription units, and 2) with GapR-seq, a recently published technology that uses a small DNA-binding protein as a probe for overtwist associated with +sc. Surprisingly, these +sc associated Top2 cleavage sites occur within 1 min and disappear rapidly (<10 minutes). Lastly, the authors show that these transient Top2 cleavage sites cannot be induced by altering transcription alone, as changing transcription by growth in galactose vs raffinose does not impact Top2 cleavage patterns, except upon osmotic treatment.

The data presented by Gittens et al is abundant, well-analyzed, and thought-provoking. That said, the central conclusion - namely, that chromatin buffers +sc and prevents Top2 cleavage of these supercoils - is insufficiently supported due to needed additional studies and by technical considerations (see below). In my opinion, the authors have an interesting study of Top2 activity during steady state and osmotic shock but it is unclear 1) if the osmolarity-induced Top2 cleavage is due to nucleosome rearrangement/buffering or via an alternative mechanism (e.g. slowdown of Top2 religation), and 2) how nucleosome rearrangement and +sc buffering is distinguished. The ms will be significantly strengthened if the authors can show a definitive link between nucleosome rearrangement/buffering and Top2 activity.

Major issues:

1) The authors show that osmotic shock reveals new Top2 cleavage sites that are associated with sites of +sc. The authors posit that osmotic shock perturbs chromatin, but the data to support this model is based on analysis of known TF binding sites or nucleosome positions (Fig. 3). An assay that directly shows chromatin perturbation (e.g. differential nucleosome positioning or DNA accessibility by histone ChIP-seq, MNase sequencing, or DNase I sequencing) at these loci during osmotic shock is necessary to support the core hypothesis. See also point #3 regarding differentiating +sc buffering vs nucleosome repositioning.

We thank the reviewer for these helpful comments, which are in-line with those of reviewers 1 and 2. As also described in our response to reviewer 2: We have now performed MNase-seq experiments throughout our osmotic shock time course, in duplicate (Fig. 5H and S4, lines 196–205). These show subtle changes in nucleosome occupancy that are spatially and temporally correlated with the more dramatic redistribution of Top2 activity, lending support to our model that osmotic shock induces chromatin perturbation that causes increased formation of Top2 substrates at sites of transcriptional stress. Despite this, we acknowledge that defining the details of our favoured model regarding chromatin buffering of topological stress is beyond the technical capacity of current tools. Therefore we have toned down definitive claims about

chromatin buffering of topology throughout our manuscript, and clarified that it is our favoured model to explain our observations (Title change, lines 54, 168–178, 218–220, 249, 268, 282–283, 289–290, 354, 358).

An alternative possibility is that osmotic change could impact Top2 catalysis. If sorbitol treatment decreases Top2 reaction rate (e.g. by increased DNA-binding or decreasing DNA religation), that would manifest as topoisomerase “trapping” on a +sc substrate. This could explain the entirety of the results of the ms and could be potentially tested by using a drug that traps Top2 on the DNA (etoposide) and repeating the Top2-cc assay. If Top2 trapping occurs, there should be minimal changes in Top2-cc between the \pm sorbitol conditions. This issue is compounded by the fact that the authors did not benchmark their novel Top2-cc assay against the previous literature (see #3 below). An additional, perhaps unlikely possibility is that there could be Top2 recruiting factor(s) that recognize these regions upon osmotic shock.

We thank the reviewer for these thoughtful comments. Whilst we consider this an interesting alternative hypothesis, we deem it unlikely because if sorbitol reduced Top2 re-ligation rate, it is unclear to us why this would occur differentially across the genome, with a clear preference for regions of transcription-induced stress. Nevertheless, to formally exclude this possibility, we have performed an experiment with etoposide, as suggested by the reviewer (Fig. S6I–K, lines 260–266). Notably, etoposide treatment failed to reproduce the hotspots of Top2 activity induced by osmotic shock, and in fact slightly suppressed these effects—arguing against the alternative hypothesis that sorbitol stabilises Top2ccs, and instead support our preferred interpretation that the observed hotspots of CC-seq signal represent Top2 activity on newly available substrates.

Despite this, we accept that with current tools we are unable to precisely elucidate the specific details of these mechanisms—which indeed may be multifaceted/multilayered—and as such we have toned down definitive claims regarding chromatin buffering of superhelical stress throughout our manuscript.

2) The Top2-cc profile in this ms differs significantly from previously published yeast etoposide-treated Top2-cc profiles, including a dataset published by the authors themselves. In Gittens et al 2019, etoposide-induced Top2-ccs occurred in divergently transcribed regions and immediately upstream of transcriptional start sites. Why is the native Top2 dataset in this work so different than the previously published etoposide-treated dataset? Is there any reason why Top2 cleavage should have a strong +sc bias given that the enzyme is able to relax both positive and negative supercoils in vitro (Goto and Wang, 1982) and the etoposide-trapped Top2 cleavage sites suggest a bias towards negative supercoiling (Gittens et al 2019)? Is it possible that the “transient” Top2-ccs being captured here are the difficult to resolve Top2-ccs, i.e. peaks indicate Top2 difficulty or “persistent” Top2-ccs? One could then view the height of each Top2-cc peak in this work not as the frequency that a site is cut by Top2, but the religation rate of each individual sequence. Could this interpretation explain the minimal correlation between Top2 cleavage and transcription in the absence of osmotic stress and the fact that Top2 cleavage occurs semi-uniformly in genomes? This could be potentially tested in vitro using

purified Top2. It is important to distinguish between these models and to provide a comparison between this and the previous Top2-cc work to understand why these datasets provide such different portraits of Top2 function.

Many thanks for this feedback, to which we believe requires a detailed response. Firstly, maps produced by CC-seq v1 (+VP16) and CC-seq v2 (untreated) actually do contain many similarities—including enrichment of signals in all types of IGRs—especially divergent and tandem. We acknowledge that these points may have been unclear in our initial submission. We have now addressed this by introducing a new supplementary figure that compares untreated and VP16-treated v1 maps with untreated v2 maps (Fig. S2, lines 9–103). Despite the similarities between the data sets, notably the dynamic range of untreated CC-seq-v2 data is higher than even VP16-treated data (FIG S2H), implying a higher signal-to-noise ratio.

From our new supplementary figure and updated main text (lines 91–103), we hope that the reader can better appreciate that whilst there are significant similarities between the v1 (VP16) and v2 (untreated) maps (Pearson R = 0.65 at 100bp resolution)—as would be expected—there are also differences, both at broad scales (Fig S2A) and finer scales (Fig SB–E). There are several aspects that could explain these differences. The prior v1 data were generated in asynchronous multidrug-sensitive *pdr1*Δ yeast treated for 4 hours with etoposide, whereas the v2 data in this were generated in VP16-untreated G1-arrested W303 yeast. Therefore, there are likely to be significant differences to the transcriptional landscape, driven primarily by population level cell cycle differences and the DNA damage response (DDR), and to a lesser extent, perhaps strain-specific differences.

At fine scale, signals in both maps are patterned by nucleosome positioning, and primary DNA sequence features. Importantly, however, the v1 (+VP16) cleavage sites are biased by specific DNA sequence features that are associated with preferential VP16-binding to the DNA-Topoisomerase complex—most notably the strong propensity for G and C at the -3 and +3 positions respectively (Fig. 1D). Furthermore top and bottom strand CC-seq-V2 signals are more highly correlated at the 3bp-offset than CC-seq-v1 signals, again implying higher signal-to-noise and/or maps that better reflect the intrinsic cognate structure of the Top2-DSB. This is likely because etoposide is known to trap individual active sites of the Top2 homodimer, introducing non-cognate Top2ccs in a manner influenced by primary DNA-sequence at the cleavage site (as discussed in Gittens 2019).

Regarding the other important point about relative activity of Top2 on negative vs positive superhelicity we would like to clarify a few points. In the absence of osmotic shock, we only detect very weak correlations with transcription: Top2 activity is very weakly positively correlated with convergent transcription rates, and very weakly anticorrelated with divergent transcription rates (Fig. 3G). However, we really meant to highlight the fact that under normal circumstances, other factors than transcription rates likely predominate in patterning Top2 activity—in particular the increased DNA accessibility at nucleosome-free regions in promoters. Therefore, we did not intend to make any strong claims about Top2 preference for positive vs negative supercoils, and have toned this down in our manuscript (lines 160, 251, and 342-361).

3) The authors posit that +sc is buffered by nucleosomes and utilize the GapR-seq technique to report evidence of +sc in cycling cells. If osmotic shock perturbs chromatin, the authors should distinguish between nucleosome rearrangement and +sc buffering. If +sc buffering occurs, then the GapR-seq pattern should overlap with the Top2 cleavage pattern during osmotic shock, but in 6E, the GapR-seq patterns are more limited while the Top2 cleavage sites extend significantly past where +sc is captured. This discrepancy should be explained. An alternative possibility is that osmotic shock generally dislodges nucleosomes (independent of +sc) increasing DNA accessibility to Top2. Nucleosome disruption will occur more frequently at transcription termination sites (where +sc accumulates) as terminators are nucleosome-bound while promoters are more nucleosome-free (Chereji et al, 2017).

Thanks for these comments. We would firstly like to highlight the limitations of currently available methods to map DNA topology—also outlined above in response to Referees 1 and 2: Currently there are two methods used to map DNA *twist*—bTMP-seq and GAPR-seq, which aim to map negative and positive superhelical twist, respectively. The expectation should be that these methods should anticorrelate across the genome, but in fact this is not the case—rather, they positively correlate with a moderate Pearson R of 0.32. This inconsistency is likely to be explained by aspects such as chromatin/DNA accessibility that are already known to influence ChIP-seq maps, coupled with secondary complications arising from steps that are technically necessary for each method (cell permeabilization/spheroplasting for bTMP-seq, and 6 hour GapR-overexpression for GapR-seq). Secondly, we highlight that DNA twist and writhe—whilst predicted to be interconvertible—may not perfectly colocalize in the genome, such that the regions of DNA over-twist mapped by GapR-seq may not perfectly overlap writhed substrates that Top2 may target upon osmotic shock.

Regarding the final suggestion that osmotic shock might generally dislodge nucleosomes, and that this would lead to increases in Top2 activity at transcription termination sites (TTS) that are more nucleosome-rich, than promoters that are less nucleosome rich, we find this unlikely. For example, we do not detect significant increases in Top2 activity immediately downstream of transcription start sites (TSS), which are the most nucleosome-rich regions of the genome on average (higher nucleosome occupancy than terminal regions of genes/TTSs).

4) The authors provide no model for the semi-uniform Top2 cleavage patterns observed in unstressed growth. Are the Top2 sites merely the sites that are nucleosome-free? What if anything explains Top2 cleavage patterns? If the model that only persistent Top2 cleavage sites have been captured is correct, the data could be interpreted that in unstressed conditions transcription-associated Top2 activity is too fast to capture, with osmotic shock slowing activity and revealing Top2 association with transcription. This could also explain the results at the galactose locus, in which Top2 activity at GAL is only observed in galactose + osmotic shock.

We agree that a major factor that patterns distributions of Top2 activity is DNA accessibility, largely (though not entirely) regardless of local transcriptional activity. Secondly, supported by our observations made upon osmotic shock, we favour the view that at least some superhelical stress is additionally hidden from Top2 due to absorption within the chromatin itself. Altogether,

at a broad scale, we argue that this leads to the semi-uniform Top2 cleavage patterns that we observe in unshocked conditions.

As detailed above, we disfavour the hypothesis that we are only able to capture a subset of Top2ccs with a longer lifetime. Nevertheless, as requested and described above (response to point #1), to test this idea, we have now performed an experiment with etoposide, which inhibits Top2 re-ligation rates throughout the genome (Fig. S6I–K, lines 260–266). However, such treatment did not reproduce the hotspots of Top2 activity formed by osmotic shock, and in fact slightly suppressed such effects. Collectively these data argue that the effect we see upon osmotic shock is not due to a slowing of Top2 activity, as proposed.

Minor points:

1) Is top2-td the Top2 with the degron tag?

Yes, this is detailed in our strain Table S1.

2) I found many of the figures difficult to read. The colors and lines are not intense enough to distinguish easily, and the fonts in many cases are too small to be easily legible.

We have now adjusted our line colours and widths throughout the text to improve their distinction.

AUTHOR RESPONSE to **NCOMMS-24-03234A**:
Reviewers comments in **BLUE**; Author responses in **BLACK**

REVIEWER COMMENTS

Reviewer #1 (Remarks to the Author):

I appreciate the effort from the authors in addressing the comments in the first review. I am convinced by some of the new results presented, and I acknowledge the technical advance that this study constitutes, as well as the interesting observations presented. However, I still bear some doubts as to whether the proposed model constitutes the most likely explanation for their observations, or whether alternative possibilities (as suggested in the first review) should be considered in more depth. Please note that, although toned down, the main message of the manuscript is still that disrupting the buffering capacity of chromatin results in increased topological stress and increased Top2 activity, so this needs to be proven.

We appreciate the positive comments about our study. We have also taken onboard the considerations raised and have further toned down the central concept when describing our results (e.g. Paragraph starting Line 168). Whilst we still favour idea that disruption of chromatin buffering can be the explanation for our induced Top2 activity—and disagree that large-scale loss of histones would be required for this to happen in vivo—we have reorganised and expanded our discussion to include alternative explanations for our findings, and have identified future work that would be required to further elucidate the mechanisms at play (Lines 313 to 362).

As indicated in the first review, strong changes in chromatin structure would need to be observed to explain a loss in the buffering capacity of chromatin as the cause of the increase in Top2 activity. This has not been observed, as the osmotic shock results in only minor changes in genome-wide MNase sensitivity. It is true that the authors observe a correlation between changes in MNase-seq and CC-seq v2, but this may just reflect the preference of Top2 for nucleosome-free regions, and not, as put forward by the authors, that the changes in chromatin result in increased accumulation of topological stress. The data obtained with the chromatin assembly mutants could support the model proposed, provided that mutants do show a correlation between Top2 activity and transcription. This should be specifically analyzed and included in the manuscript. I do not agree that acute experiments are needed in this regard. If these mutants have stable disrupted chromatin, even if subtle (as transiently observed upon osmotic shock), topological stress and Top2 activity should be increased, if the model proposed by the authors is correct. I consider this essential for the publication of the manuscript.

We thank the reviewer for their critical evaluation of our study. Whilst the chromatin mutants lend support towards our model, they are preliminary and constitute only the initial phase of a much wider study, which is why we included them only in our rebuttal. We consider that transcriptomics in matched cells, controlling for any cell cycle changes, potentially with matched MNase-seq, and possibly also characterising effects (or not) on sensitivity to osmotic perturbation are all important aspects to such data, and are thus clearly follow-on work that goes beyond the scope of this current study.

Regarding the measurement of DNA supercoiling, I agree that it may be challenging to use GapR-seq or bTMP-seq. However, I do not understand why, as suggested in the first review, the authors do not measure supercoiling directly in a yeast mini-chromosome. This would be an unambiguous way to demonstrate that osmotic stress induces supercoiling, and even distinguishing between positive/negative supercoiling and determining the requirement for transcription could be measured relatively easily.

We agree that use of a mini chromosome might be an interesting way to further probe mechanism in the future.

However, we strongly feel that setting up such an assay at this stage is beyond the current scope of the study. In particular, alongside the time needed to implement this assay within our lab, we would need to generate large constructs with arrays of convergent and/or inducible genes, and even if achieved, it remains likely that super-helical stress (and/or buffering) manifests differently on a circular molecule than it does on large linear chromosomes. For these reasons, we feel strongly that they are beyond the scope of this study.

Finally, regarding the correlation with nascent transcription, I do understand the technical difficulty, but at least the authors should have aimed to have condition-matched datasets for correlation. Osmotic stress will change the transcriptional landscape even at those short timepoints, so I do not think it is correct to correlate unperturbed RNA-seq data with CC-seq v2 after osmotic shock. If this is not what authors do, I apologize. In any case, I agree that one would expect, if something, a stronger correlation, so, in contrast to the above points, I do not consider this to be essential.

MJN: We maintain that such experiments would be technically challenging with current methodologies over such short timeframes and, given the already observed correlation, we agree with the reviewer and editor that they are not essential.

Reviewer #2 (Remarks to the Author):

The authors have addressed most of my points.

The manuscript has improved, and the claims and statements in the manuscript are now more balanced.

I would have liked to see some genome-wide transcription inhibition experiments, but I see why the authors are reluctant to go in that direction.

I particularly appreciated how the authors have addressed points from the other reviewers with solid experimental work.

I recommend acceptance.

Stefano G Manzo

We are grateful to the reviewer for their constructive comments that have helped improve the manuscript, and for their additional positive message of support.

Reviewer #3 (Remarks to the Author):

Gittens et al have resubmitted a revised version of their manuscript describing a new version of Top2 activity mapping showing how activity is sensitive to osmotic state. The novel Top2 mapping technique and the observed rapid changes in Top2 activity are important advances in the field. I commend the authors for the extensive set of experiments in the revised manuscript. The new data and the clear rewrite address all my earlier concerns.

As above, we are grateful to the reviewer for their constructive comments that have helped improve the manuscript, and for their additional positive message of support for the revised version.